# A Comparative Evaluation of Snowflake Particle Size and Shape Estimation Techniques used by the Precipitation Imaging Package (PIP), Multi-Angle Snowflake Camera (MASC), and Two-Dimensional Video Disdrometer (2DVD)

Charles N. Helms[1,2], S. Joseph Munchak[1], Ali Tokay[1,3], and Claire Pettersen[4]

[1]Mesoscale Atmospheric Processes Laboratory, NASA Goddard Space Flight Center, Greenbelt, MD, USA
[2]NASA Postdoctoral Program—Universities Space Research Association, Columbia, MD, USA
[3]Joint Center for Earth Systems Technology, University of Maryland, Baltimore County, Baltimore, MD, USA
[4]Climate and Space Sciences and Engineering Department, University of Michigan, Ann Arbor, MI, USA

**Correspondence:** Charles N. Helms (charles.n.helms@nasa.gov)

**Abstract.** Measurements of snowflake particle size and shape are important for studying the snow microphysics. While a number of instruments exist that are designed to measure these important parameters, this study focuses on the measurement techniques of three digital video disdrometers: the Precipitation Imaging Package (PIP), the Multi-Angle Snowflake Camera (MASC) and the Two-Dimensional Video Disdrometer (2DVD). To gain a better understanding of the relative strengths and weaknesses of these instruments and to provide a foundation upon which comparisons can be made between studies using data from different instruments, we perform a comparative analysis of the measurement algorithms employed by each of the three instruments by applying the algorithms to snowflake images captured by PIP during the ICEP-POP 2018 field campaign.

Our analysis primarily focuses on the measurements of area, equivalent diameter, and aspect ratio. Our findings indicate that area and equi-area diameter measurements using the 2DVD camera setup have the greatest accuracy of the three instruments due to its resilience to motion blurring effects. Both PIP and MASC use shape-fitting algorithms to measure aspect ratio. While our analysis of the MASC aspect ratio suggests that the measurements are reliable, our findings indicate that both the ellipse and rectangle aspect ratios produced by PIP under-performed considerably due to the shortcomings of the PIP shape-fitting techniques. That said, we also demonstrate that reliable measurements of aspect ratio can be retrieved from PIP by reprocessing the PIP images using either the MASC ellipse-fitting algorithm or a tensor-based ellipse-fitting algorithm. Because of differences in instrument design, 2DVD produces measurements of particle horizontal and vertical extent rather than length and width. Furthermore, the 2DVD measurements of particle horizontal extent can be contaminated by horizontal particle motion. Our findings indicate that, although the correction technique used to remove the horizontal motion contamination performs remarkably well with snowflakes despite being designed for use with rain drops, the 2DVD measurements of particle horizontal extent are potentially unreliable.

## 1 Introduction

Digital video disdrometers have been in use since the 1990's and there exist several instruments designed around using digital cameras to observe precipitation properties. The measurements from these instruments are not only critical for understanding precipitation microphysics but are also critical for developing algorithms to retrieve microphysical properties using radar and passive microwave sensor data. Due to the variety of instruments available, issues can arise when making comparisons between studies and algorithms that are based on data from different instruments. To this end, the present study will perform a comparison of the particle area, equivalent diameter, and aspect ratio measurements produced by three common ground-based digital video disdrometers: the Precipitation Imaging Package (PIP; Pettersen et al., 2020, 2021), the Multi-Angle Snowflake Camera (MASC; Garrett et al., 2012), and the Two-Dimensional Video Disdrometer (2DVD; Kruger and Krajewski, 2002; Schönhuber et al., 2007, 2008).

Studies invoking snowflake particle size frequently rely on either the maximum dimension length of a particle (e.g., Locatelli and Hobbs, 1974; Heymsfield and Westbrook, 2010), typically defined as either the major-axis length of a fitted ellipse or as the diameter of the smallest circumscribing circle, or on the equivalent diameter (e.g., Han and Braun, 2021; Pettersen et al., 2020, 2021), defined as the diameter of a circle or sphere with equal projected area (for the equi-area diameter) or volume (for the equi-volume diameter) to a particle. 2DVD combines orthogonal measurements of a particle to estimate the particle volume, from which the equi-volume diameter is computed while both PIP and MASC compute the equi-area diameter from their respective area measurements, both of which are determined by counting the number of pixels within the particle. That said, there have been efforts to use the multiple cameras mounted by MASC to compute particle volume (Leinonen et al., 2021).

In the case of Han and Braun (2021), the authors characterize the global three-dimensional distribution of precipitation mean particle sizes using satellite radar data; they used a form of the equivalent diameter as their metric of particle size, specifically the mass-weighted mean liquid-water-equivalent diameter. As with many studies that invoke particle size distributions, their choice of metric came down to the choice of metric used in the original particle size distribution parameterization. For Han and Braun (2021), the authors used data produced following Grecu et al. (2016), which used the particle size distribution parameterization described in Testud et al. (2001). Testud et al. (2001) chose to base their parameterization on equivalent diameter, specifically the volume-weighted mean liquid-water-equivalent diameter, for two key reasons: the authors argue that this variable has less physical ambiguity than other metrics and is relatively easy to compute.

The equivalent diameter measurements have also been used to determine a number of bulk snowflake characteristics. Pettersen et al. (2020) used PIP-observed equivalent diameter measurements to compare a number of microphysical and bulk characteristics of snow, including the liquid-water-equivalent snow rate and the bulk effective density of the snow particles, to collocated observations. Their study found that the liquid-water-equivalent snow rate compared very well to both established retrievals as well as observations. Using these bulk characteristics, Pettersen et al. (2021) was able to discriminate between different precipitation phases using the PIP observations.

Empirical relationships between snowflake or ice crystal size and terminal fall speed frequently use the maximum dimension of the particle as their independent variable (e.g., Locatelli and Hobbs, 1974). The more advanced relationships also incorporate the Reynolds number, which is dependent on the particle area exposed to the flow via the Best number; in some cases this area is replaced by an area ratio that is defined as the ratio between the area exposed to the flow and the area of the circumscribing circle, which is a function of the particle maximum dimension (Heymsfield and Westbrook, 2010). In this way, the terminal fall speed relationships can depend on both the equivalent diameter, which is directly tied to particle area, and the maximum dimension of the particle. Similarly, combinations of equivalent diameter and maximum dimension have also been used to estimate particle mass (e.g., von Lerber et al., 2017).

Snowflake shape is generally characterized by the aspect ratio of the snowflakes. Korolev and Isaac (2003) found that the mean aspect ratio of in-cloud ice particles varied as a function of the particle maximum dimension for particles smaller than ∼65 $\mu$m whereas the aspect ratio of larger particles (i.e., those typical of snow) tended to vary inversely with temperature; the mean aspect ratios observed by the study were between 0.6 and 0.8 with the snowflake-sized particles tending to have mean aspect ratios closer to 0.6. To constrain radar backscatter calculations, aspect ratio is frequently prescribed, often with a mean value of 0.6 assumed (e.g., Matrosov et al., 2005), although a number of studies have performed backscatter calculations using a range of prescribed aspect ratios (e.g., Tyynelä et al., 2011; Westbrook, 2014). The reverse of this process has also been attempted: Munchak et al. (2022) used an optimal estimation technique to retrieve a number of snow microphysical properties, including aspect ratio, from a dual-frequency dual-polarization ground-based radar, albeit with mixed results.

As will be discussed in greater detail below, both PIP and MASC produce their measurements by fitting simple two-dimensional shapes (ellipses and rectangles, specifically) to two-dimensional projections of the three-dimensional snowflakes that the instruments are observing. Because snowflakes come in a large variety of shapes, especially when taking aggregate snowflakes into consideration, any attempt to use a simple shape, such as an ellipse or rectangle, to represent these particles suffers from the inherent limitation of under-representing the complexity of the snowflakes. Furthermore, the use of two-dimensional shapes to represent three-dimensional snowflakes adds an additional layer of limitations revolving around the degree to which the two-dimensional projection accurately represents the dimensions and orientations of the three-dimensional particle (e.g., Jiang et al., 2017). Despite these shortcomings, the measurement of snowflake dimensions based on shape fitting has proven to be a useful tool for studying snow microphysics and understanding the relative capabilities of these measurements is critical to their successful use in research and applications.

The goal of the present study is to evaluate the techniques used by three ground-based digital video disdrometers to determine the size and shape of precipitation particles in terms of the area, equivalent diameter, and aspect ratio. In doing so, we will provide a point of comparison for these three sensors. The next section will briefly discuss the data sets used for this study. In section 3, we will provide a detailed description of each sensor and their respective processing algorithms as they relate to the particle shape. This will be followed by a description of our analysis methods in section 4. Section 5 will present the results of our study. Section 6 will compare the results of the present study to the findings of previous studies. Finally, section 7 will present our conclusions alongside a brief summary of the study.

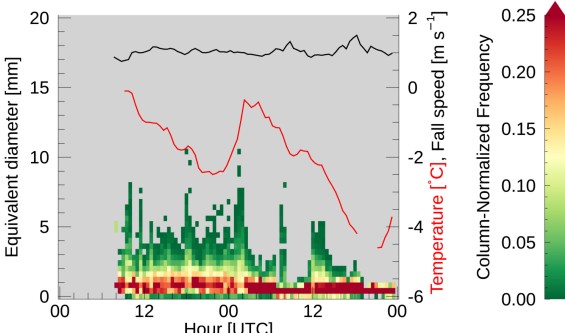

**Figure 1.** Time series of (shading) particle size distribution (in terms of equivalent diameter [mm]), (black) mean particle fall speed [m s$^{-1}$], and (red) collocated air temperature [°C] at the MHS site on 7–8 March 2018. Equivalent diameter and fall speed measurements come from the PIP data and the air temperature was measured using a Vaisala WXT520. The particle size distribution time series histogram has been normalized in the y direction.

## 2   Data

The data sets used to perform the analyses central to this study were collected during the 2018 International Collaborative Experiment for PyeongChang Olympic and Paralympics (ICE-POP;  Lee and Kim, 2019; Lim et al., 2020; Tapiador et al., 2021). ICE-POP, which took place between 1 November 2017 and 17 March 2018, was an international collaboration between a number of programs and agencies, including the NASA Global Precipitation Mission (GPM) Ground Validation program and the Korean Meteorological Administration (KMA). The field campaign was designed to study winter precipitation in complex terrain and involved a number of measurement sites featuring a mixture of in-situ and remote-sensing instrumentation. Of interest to the present study is the May Hills supersite (MHS; 37.6652°N, 128.6996°E, 789 m) as this site has collocated PIP, MASC, and 2DVD instruments and experienced frequent snow events. This study will focus on data collected during a snow event that took place on 7–8 March 2018. This event was selected as it contains both aggregate snow particles, which provide a large variety of shapes (and, therefore, aspect ratios), and lump graupel particles, which tend to be relatively spherical with high aspect ratios. Although the general presence of these habits were primarily identified through visual inspection of the PIP data, further support of their presence can be found by examining the time series of the particle size distribution, particle fall speed, and air temperature, as depicted in Fig. 1. Periods when aggregates are present can be identified by the larger equivalent diameters and slower fall speeds and periods with lump graupel can be identified by the smaller equivalent diameters and faster fall speeds; the lump graupel can be discerned from liquid precipitation based on the below freezing temperatures, which extend over a deep layer according to a nearby thermodynamic sounding (not shown). Data collected during a snow event on 9 January 2018 and 22 January 2018 were also examined, but are not included here as their inclusion did not produce any notable changes in the results.

## 3 Instruments

This study will evaluate the measurement techniques of three video disdrometers: PIP, MASC, and 2DVD. The instruments and their processing algorithms, as they relate to measuring the shape and orientation of snow particles, will be covered in this section.

### 3.1 Precipitation Imaging Package (PIP)

PIP (Pettersen et al., 2020, 2021) is a video disdrometer developed by NASA as a successor to the Snowflake Video Imager (SVI; Newman et al., 2009). PIP is composed of two parts: a high-speed video camera and a light source to backlight the precipitation particles that pass through the open sampling volume. The camera, of the 'charge-coupled device' (CCD) variety and located 2 m from the light source, continuously records images at a rate of 380 frames per second as 640 by 480 pixel images with a pixel size of 0.1 mm by 0.1 mm using a focal plane located 1.33 m away from the camera. In order to reduce the data rate produced by the high-speed camera, the PIP images are compressed before being processed, resulting in an effective pixel size of 0.1 mm in the horizontal by 0.2 mm in the vertical. The present study uses the ICE-POP PIP data produced using version 1403 of the PIP processing code.

The PIP processing algorithms are written using the IMAQ Visual software package (National Instruments, 2000, 2003), a software package designed for use in performing automated quality control of manufacturing processes. Of particularly interest to the present study are the methods the algorithm uses to compute the dimensions of the precipitation particle. While particle area is determined by counting the number of pixels within the particle, the particle dimensions are determined by fitting an ellipse or a rectangle to the particle as viewed by PIP, which sees the two-dimensional projection of the particle, after the PIP image has been compressed to reduce data rates. The IMAQ software package performs both the ellipse and rectangle fitting by constructing an ellipse or rectangle that has an equal area and an equal perimeter to the target particle; a derivation of the equations used to make these fits are presented in Appendix A to supplement the limited information given in the IMAQ Vision manual. According to the 2003 IMAQ Vision software manual (National Instruments, 2003), the IMAQ Vision software package computes the particle perimeter by subsampling the boundary points to produce a smoother representation of the perimeter; for this purpose, the boundary points are located at the corners of the pixels that make up the particle perimeter. Our interpretation of this is that a corner pixel (i.e., a particle pixel sharing exactly two adjacent sides with other particle pixels) will contribute only its diagonal length to the perimeter rather than the length of the exterior two sides. How this subsampling behaves for a particle pixel that only shares a single side with other particle pixels is unclear. Once the PIP-fitted shape has been determined, the long and short axes of the fitted shapes are then used as the length and width of the particle. Specifically, the ellipse fit produces the variables 'Ellip_Maj' and 'Ellip_Min' for the length and width, respectively, while the rectangle fit produces the variables 'Rec_BS' and 'Rec_SS' for the length and width, respectively. Note, in both the ellipse and rectangle fits, the perimeter and area are defined such that any holes in the particle are not taken into account. Because of this, the PIP-determined area and perimeter of a particle with no holes will be the same as an otherwise-identical particle with a hole when determining the shape fits.

## 3.2  Multi-Angle Snowflake Camera (MASC)

MASC (Garrett et al., 2012) is another video disdrometer and was developed at the University of Utah; the setup described below is that used during the ICE-POP field campaign. Unlike PIP, MASC employs three cameras and only records images when triggered by a particle falling through the field of view of a pair of near-infrared sensors. The near-infrared sensors are also used to trigger a flash, which illuminates the particle from the side facing the cameras. The three cameras, also of the CCD variety and separated from one another by $36°$, are mounted looking into an enclosed sampling volume, open only at the top and bottom, and have a common focal point 10 cm away from each lens. Typically, the camera system is artificially restricted to only being triggered once per second in order to reduce excessive flashes. The sampling volume is defined by the intersection of the 35-mm fields of view and the 10-mm depths of field of the three cameras. While the cameras have a pixel size of 9 $\mu$m by 9 $\mu$m, the system is only triggered when particles with a maximum dimension of greater than 0.1 mm are detected by the near-infrared sensors.

Although the MASC processing algorithm was originally written in MATLAB, a version written in Python also exists and it is this version of the ellipse fitting method that will be analyzed here (Shkurko et al., 2016). To simplify matching particles between the three cameras, MASC only processes images that contain a single precipitation particle and, of these, only particles that are in focus are actually analyzed. The method used to determine whether a particle is in focus is detailed in Garrett et al. (2012). Similar to PIP, MASC also fits an ellipse to the precipitation particles to determine the particle shape properties. In the Python version of the MASC algorithm, this is accomplished using the fitEllipse function within the OpenCV package (https://docs.opencv.org/3.4/, accessed 15 December 2021), which uses the 'LIN' algebraic distance ellipse-fitting algorithm described in (Fitzgibbon and Fisher, 1995) and returns both the major and minor axis lengths as well as the orientation angle of the fitted ellipse.

## 3.3  Two-Dimensional Video Disdrometer (2DVD)

2DVD (Kruger and Krajewski, 2002; Schönhuber et al., 2007) uses two orthogonally oriented horizontal line-scan cameras to take measurements of backlit precipitation particles. Since the viewing planes of the two cameras are separated by 6–7 mm, the particle fall speed can be measured via the difference in the arrival time of the particle in each viewing plane. These horizontally scanning line-scan cameras capture a series of one-dimensional images of the particle as it passes through the viewing plane. Each camera captures a one dimensional image every $18\mu$s and these one-dimensional images are then pieced together by stacking each scan on top of one another with a thickness dependent on the observed fall speed to produce a two-dimensional image of the precipitation particle for each camera. Each of the 2DVD line-scan cameras has 512 photodetectors that are calibrated to observe a 100-mm wide field of view (Kruger and Krajewski, 2002), from which we can infer that the pixel size is $\sim$0.195 mm (195 $\mu$m). Although 2DVD was designed to study rain drops, the instrument has been used to study snow as well (e.g., Huang et al., 2010).

Because the 2DVD cameras do not capture the entire shape of the particle at once, consideration must be given for how the horizontal motion impacts the perceived particle shape. For a particle with no horizontal motion, piecing together the

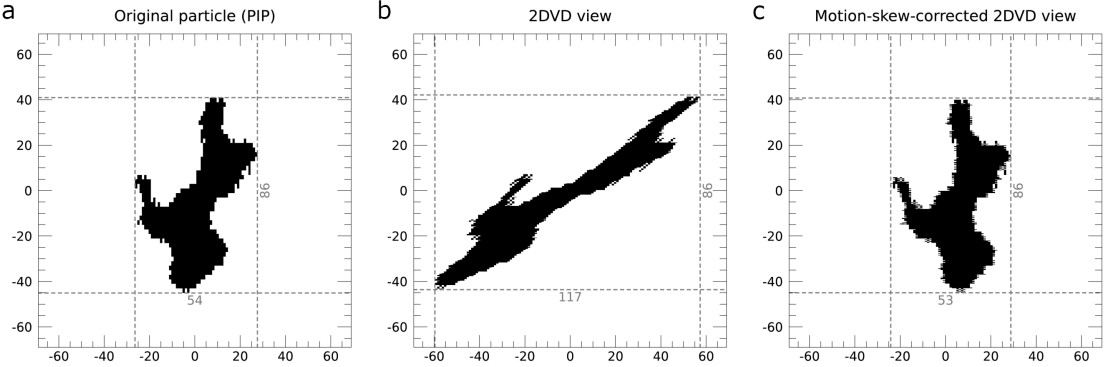

**Figure 2.** An example of the emulated 2DVD measurements and our unskewing algorithm including (a) the original particle as captured by the PIP camera, (b) the emulated 2DVD view of the particle before the skew correction, and (c) the emulated 2DVD view of the particle after applying our skew correction. The coordinates are in units of PIP pixels from the particle center. The gray dashed lines indicate the extent of the bounding boxes for each view of the particle, with the length of each bounding box edge marked on the bottom and right edges in units of PIP pixels.

one-dimensional images will result in a particle image that depicts a close approximation of the actual particle. If the particle is moving horizontally, however, then the particle will appear to have been shifted horizontally in each subsequent one-dimensional image. The end result of imaging a horizontally moving particle is that the particle will appear to have undergone a shear transformation relative to the actual particle. Figure 2a–b depicts an example of this shearing effect by sampling a PIP image as though a 2DVD instrument were measuring the particle (see section 4.2).

The image skewing produced by the horizontal movement of particles, as evident in Fig. 2b, can invalidate certain 2DVD measurements. While an algorithm designed to remove these effects exists, the algorithm is, unfortunately, in the form of a proprietary software package and, as such, its exact inner workings are unclear. That said, Kruger and Krajewski (2002) state that the algorithm works by assuming the top-most and bottom-most pixels of the particle should be vertically aligned with one another. The algorithm then skews the image in the opposite direction to a sufficient degree to place the top- and bottom-most pixels into vertical alignment. An approximation of the unskewing algorithm has been applied to the particle in Fig. 2c; this approximation of the unskewing algorithm is described in section 4.2.

Unlike MASC and PIP, the equivalent diameter that is actually recorded in the 2DVD data sets is an equi-volume diameter. The details of how this equi-volume diameter is computed from the raw 2DVD data can be found in Schönhuber et al. (2008). Briefly, the calculation works by assuming the shape observed in each pair of scan lines, captured by the orthogonal pair of line-scan cameras, is that of an elliptical cylinder whose major and minor axes are entirely captured by the pair of line-scan cameras. The volumes of these elliptical cylinders are then summed to determine the total volume of the particle. While this method is reasonable for approximately ellipsoidal particles (e.g., raindrops or hailstones), the complex surface structures present in snowflakes and snowflake aggregates can invalidate the volume calculations.

## 4 Method

The ideal method for comparing the measurements made of snowflakes by each of these three instruments would be to have all three instruments measure the same particles simultaneously. Due to the construction of each of these instruments, however, this is not physically possible. Instead, the algorithms used by each instrument to determine particle shape will be analyzed by emulating each instrument's analysis algorithms using PIP-recorded images as an input. PIP images are used as the basis of the measurement emulation rather than MASC images because the PIP captures considerably more particles in its sampling volume than does MASC.

The PIP images used for this study come in the form of Audio Video Interleave (AVI) video files. PIP produces these video files every 10 minutes and each video file contains the first 2000 frames in which PIP identified a particle during that 10 minute period. If fewer than 2000 frames contained particles during a given 10 minute period, the video file will contain fewer than 2000 frames. A description of how we emulate the MASC and 2DVD observations is provided in sections 4.1 and 4.2 and an overview of the method by which we process the PIP AVI video files in order to apply our emulated algorithms is presented in Appendix B. Additionally, we will test and evaluate an alternative ellipse-fitting method, which employs a mass tensor and is hereafter referred to as the tensor method, for retrieving particle length and width from the PIP images; this method is described in section 4.3.

Ideally, we would include all PIP images from the 7–8 March 2018 period in our study rather than just those images that are included in the AVI files. Since the raw PIP image files are archived and contain every frame with precipitation, this is theoretically possible. That said, due to the very high data rates involved with the PIP data, the processing code would need to be highly optimized or use specialized software packages in order to process even a couple days of the raw data in a reasonable time frame. Thankfully, the AVI files contain a sufficiently large collection of precipitation particles to enable us to perform our study.

For the emulation of both MASC and 2DVD as well as the implementation of the tensor method, the particle image pixels have values of either zero (non-particle pixel) or one (particle pixel). Any pixel that does not have the background value (i.e., white) is included as part of the particle. Particles in the AVI files are matched to the PIP measurements by using a region grow approach, whereby a coherent region is grown from a single point by checking its eight neighboring pixels for inclusion in the region and then repeating the process with each newly included pixel. The region grow approach is initially attempted at the PIP-determined particle centroid. For some particles with either large concave regions or holes, the PIP-measured particle center can correspond to a non-particle pixel. In these instances, an attempt is made to locate a new starting pixel for the region grow technique by searching the following (x,y) coordinate offsets in the given order: (-1,0), (1,0), (0,-1), (-1,-1), (1,-1), (0,1), (-1,1), (1,1). If a particle pixel is not located at any of these center offset positions, the process is repeated after multiplying the offsets by a factor of two; in cases where a particle pixel remains elusive, the process is repeated with multiples of three and then four. If no particle pixels are located after checking these 32 offset positions, the particle is considered invalid and is ignored. Additionally, only particles that appear both in the particle tables (i.e., the PIP files ending in '_a_p.dat') and in one

of the two velocity tables (i.e., the PIP files ending in either '_a_v_1.dat' or '_a_v_2.dat') are considered here as the variables contained in these files are necessary for performing the emulations.

## 4.1 MASC emulation

Our emulation of the MASC processing algorithm is based on the algorithm used for the U.S. Department of Energy Atmospheric Radiation Measurement (ARM) program MASC instruments (Stuefer and Bailey, 2016; Shkurko et al., 2016). Because both the PIP and MASC ultimately perform their measurements using two-dimensional images, no special processing is performed to prepare the images before emulating the MASC measurements. The emulation of the actual measurements is performed by using the same ellipse fitting function as is used by the ARM MASC instrument algorithm, specifically the fitEllipse function from the OpenCV Python package (see section 3.2). Although the MASC cameras have a considerably higher resolution than the PIP camera, it is worth noting that the effects of the camera resolution difference are expected to be negligible in terms of shape fitting. One exception to this is that the fitEllipse function effectively requires at least five pixels within the target particle; for the actual MASC measurements, this represents a particle far smaller than the threshold minimum measured size of 0.1 mm. In the case of the emulated MASC, a five PIP-pixel particle would have an area of 0.5 mm$^2$ and a maximum dimension of at least 0.3 mm as the PIP pixels are calibrated to be 0.1 mm of each side. To avoid having particles that can only be analyzed by a subset of the measurement algorithms, our comparisons here will only consider particles which have an equivalent area of at least 0.5 mm$^2$.

## 4.2 2DVD emulation

The 2DVD instrument captures a series of one-dimensional images as a particle falls through the observing volume. As such, the PIP images must be resampled in a way that mimics the 2DVD instrument itself. To this end, we use snapshots of the PIP-observed particles and replicate the horizontal and vertical motions of the particles, as measured by PIP, using bilinear interpolation. The vertical motion is replicated by shifting the vertical coordinates of the bilinear interpolation upward by an amount equal to the particle fall speed divided by the camera observation frequency; for 2DVD, the camera observation frequency is 68200 Hz. The horizontal motion is replicated by shifting the particle snapshot pixel locations horizontally in the direction of particle motion by an amount equal to the distance traveled by the particle during the elapsed time of the emulated observations, to the nearest 0.1 mm. This elapsed time is equal to the product of the inverse of the camera observation frequency and the number of times the particle has been sampled in the vertical up until that point (i.e., the number of emulated 2DVD line scans performed). When performing the interpolations, only the interpolated pixels with a value of one are considered part of the emulated 2DVD observation.

An example of an emulated 2DVD observation is depicted in Fig. 2 with the original, PIP-observed, particle image depicted in Fig. 2a and the emulated 2DVD particle image depicted in Fig. 2b. Note the considerable skewing that has occurred in the emulated 2DVD image relative to the PIP image due to the horizontal movement of the particle towards the left as it passed through the virtual line scan camera. As previously mentioned, 2DVD applies an unskewing algorithm to correct for the distortions introduced by the horizontal motion of the particle. Because this algorithm is part of a proprietary software package

and not openly available, we have chosen to approximate the effects of the algorithm based on the conceptual description provided by Kruger and Krajewski (2002).

Recall, from section 3.3, that the 2DVD unskewing algorithm works by assuming that the top-most and bottom-most points on an observed particle are actually vertically aligned with one another (Kruger and Krajewski, 2002). In the event of multiple particle pixels appearing on the top-most or bottom-most scan line, it is unclear how the actual algorithm selects the top-most or bottom-most point of the particle. To this end, we have decided to define the top-most and bottom-most points of the particle as the mean horizontal location of the particle pixels on the top-most and bottom-most scan lines, respectively.

The actual unskewing process in our algorithm works by linearly interpolating the horizontal offset, as implied by the motion-skewed locations of the top-most and bottom-most points of the particle, to the vertical position of each scan line. Each scan line is then horizontally shifted by the appropriate offset via another linear interpolation. The value of the interpolated pixels are then rounded to either zero, indicating a non-particle pixel, or one, indicating a particle pixel. Figure 2c depicts the result of applying the unskewing algorithm; note the slight differences in the apparent particle orientation between the original image (Fig. 2a) and the unskewed image (Fig. 2c). We will examine the effects of these differences on the accuracy of the 2DVD bounding-box measurements in section 5.3.

### 4.3 Tensor method

To provide an additional point of comparison for the particle measurements, we also implemented an alternative ellipse fitting strategy, referred to here as the tensor method, that uses the method implemented in the fit_ellipse program in the Coyote IDL Library (accessible at http://www.idlcoyote.com, accessed 27 March 2022). This method works by computing the eigenvalues, $e_1$ and $e_2$, of a two-by-two mass distribution tensor matrix, which is defined as

$$\tau = \begin{bmatrix} \overline{(\Delta y)^2} & \overline{\Delta x \Delta y} \\ \overline{\Delta x \Delta y} & \overline{(\Delta y)^2} \end{bmatrix}, \tag{1}$$

where $\Delta x$ and $\Delta y$ are the distances from the particle centroid in the horizontal and vertical directions, respectively, and the overbars indicate averaging. The semimajor and semiminor axes, $a$ and $b$, respectively, are then calculated from the eigenvalues via

$$a = 2\sqrt{e_1} \quad \text{and} \quad b = 2\sqrt{e_2}. \tag{2}$$

The major and minor axes of the fitted ellipse can then be obtained simply by doubling the semimajor and semiminor axes. Finally, in the interest of completeness, the tensor-fitted ellipse orientation, measured counterclockwise from the positive $x$ axis, is computed as $\arctan(e_2/e_1) - 90°$. As will be discussed in section 5.2, the tensor method produces similar, although not identical, fits to that of the emulated MASC method.

 **5 Results**

**5.1 Area and equivalent diameter**

Here we will examine the theoretical accuracy of the MASC, PIP, and 2DVD area and equivalent diameter measurements in terms of the susceptibility of each instrument to motion blurring, which is a function of the pixel size, exposure time (i.e., shutter speed), and particle vertical and horizontal movement speed. For both PIP and MASC, the equivalent diameter is determined from the area of the two-dimensional projection of the particle. As discussed in section 3.3, 2DVD, however, reports an equi-volume diameter rather than an equi-area diameter by summing the volumes of stacked elliptical cylinders. For simplicity, we will only consider the effects of motion blur on the two-dimensional area and equi-area diameter from a single camera for 2DVD rather than the estimated volume and equi-volume diameter that the 2DVD software computes. Additionally, there are a number of factors that are not being considered in the present study. We have not included the effects of image thresholding on the area measurements nor have we considered the effects of each camera's point spread function, which describes how light is distributed amongst each of the imaging pixels of the camera sensor. We also do not consider the effects of the circle of confusion (i.e., the blurring of out-of-focus particles) as this will only be an issue if the particle is located outside of the instrument's depth of field.

To understand the expected impacts of motion blurring on area and equivalent diameter, we perform a statistical analysis of simulated particle motion blurring. To simulate the particle motion blurring, we generate 196 one-dimensional particles, whose lengths are uniformly distributed between 0.5 mm and 20 mm, and then add the expected motion blurring for a particle moving at 1 m s$^{-1}$, 4 m s$^{-1}$, and 10 m s$^{-1}$ in the direction of the particle's long axis for each instrument. The expected motion blurring is computed by multiplying the particle velocity by the camera exposure time. The particle velocities are chosen as follows (e.g., Locatelli and Hobbs, 1974; Garrett et al., 2012; Vázquez-Martín et al., 2021): 1 m s$^{-1}$ represents a relatively fast falling snow particle, 4 m s$^{-1}$ represents an excessively fast fall speed for snow particles, and 10 m s$^{-1}$ represents relatively fast horizontal motion (albeit below the U.S. National Weather Service wind threshold for blizzard conditions, $\sim$15 m s$^{-1}$). The motion-blurred particles are then randomly moved a fractional distance between zero and two pixel lengths before being pixelated as per the pixel sizes of each instrument. In the case of PIP, the image compression is also taken into account. This process is repeated 100000 times in total for each particle size and each fall speed. For computational simplicity, these calculations assume that a pixel is part of the particle if it has at least 50% coverage and that the motion is not oblique to the pixels (i.e., only completely vertical or horizontal motion is considered).

The MASC setup used during ICE-POP had a pixel size of 33.5 $\mu$m and an exposure time of 40 $\mu$s or 1/25000th of a second (Jacopo Grazioli, 2021, personal communication). With this exposure time, a single point on an arbitrary precipitation particle falling directly downwards at 1 m s$^{-1}$ would appear as a 40 $\mu$m long vertical line. With a pixel size of 33.5 $\mu$m, there would be minor motion blurring as a single point on the particle would contribute to at least two pixels and possibly three, although the contribution to two of the pixels in the three-pixel case would be minor. Our statistical analysis indicates an average 1.19 extra particle pixels are added, which matches the expected value that can be calculated from the motion and pixel size. Note, the actual number of pixels added in any instance will always be a whole number. For a 4 m s$^{-1}$ fall speed, which would be an

extremely fast fall speed for a snowflake, the point would appear as a 160 $\mu$m line, appearing in at least five pixels (average of

4.78 extra pixels) in the ICE-POP MASC. For a particle moving horizontally at 10 m s$^{-1}$, the point appears as a 400 $\mu$m line, contributing to at least 12 pixels (average of 11.94 extra pixels), although it should be noted that the enclosed sampling volume used by MASC would make such a particle motion less likely than it would be with an open sampling volume.

PIP has an exposure time of 28 $\mu$s or $\sim$1/35700th of a second. With this exposure time, a point on a particle moving at either 1 m s$^{-1}$, 4 m s$^{-1}$, or 10 m s$^{-1}$ would appear as either a 28 $\mu$m, 112 $\mu$m, or 280 $\mu$m line, respectively. Unlike MASC, however,

PIP uses image compression in the vertical and, as such, the effects of horizontal and vertical motions must be examined separately. For horizontal motion, corresponding to the pre-compression blurring effects, PIP has a pixel size of 0.1 mm (100 $\mu$m). As such, the motion blurring effects from 1 m s$^{-1}$ of motion would be negligible under our assumptions with only 0.28 pixels added on average. At 4 m s$^{-1}$ and 10 m s$^{-1}$, the motion blurring has a large impact with an average of 1.12 and 2.8 pixels added, respectively.

As mentioned previously, PIP compresses the images prior to transmission to the computer where the image analysis is performed. As part of the PIP compression algorithm, pairs of vertically adjacent pixels are averaged together, resulting in a pixel size of 0.1 mm by 0.2 mm while the image is compressed. When the image is decompressed, this averaged value is assigned to the individual pixels making up the pixel pair (L. Bliven, 2021, personal communication), producing an effective pixel size of 0.1 mm by 0.2 mm. Because the motion blurring effects apply to the pre-compression pixels rather than the

post-compression pixels, the interaction between image compression and motion blurring warrants a closer examination.

Figure 3 depicts schematic representations of how the effects of motion blurring and image compression interact for a single column of PIP pixels when viewing a 0.4-mm tall particle. As with our statistical analysis, we have arbitrarily decided that a pixel will count as being part of the particle when it appears to be at least 50% covered by the particle, taking both motion blurring and compression into account. In actuality, the assignment of an 8-bit PIP pixel to a particle is more complicated and is

described in Newman et al. (2009). The simplest example is that of a particle suspended in air (i.e., with zero fall speed), which will experience no motion blurring (Fig. 3a). The left column of pixels in the panel depicts the pre-compression view, which sees the particle as being four pixels tall (i.e., 0.4 mm) as there is no motion blurring or compression. Because a particle can be located anywhere relative to the compression algorithm's vertically paired pixels, we must examine the two extreme possible alignments to understand the range of possible effects. In the first of these possibilities, the top edge of the particle is aligned

with the top of the upper pixel in the pixel pair, depicted in the center column of pixels in the panel. This alignment results in the post-compression image also depicting a particle that is four pixels tall. In the second of these possibilities, the top edge of the particle is aligned with the top of the lower pixel in the pixel pair, depicted in the right column of pixels in the panel. This alignment results in the post-compression image depicting the particle as being six pixels tall instead of four pixels tall because the completely covered top- and bottom-most pixels of the particle are averaged with completely uncovered non-particle pixels

resulting in a post-compression pair of pixels that each appear to be 50% covered by the particle (as indicated by the number next to the dark red brace in Fig. 3a). As such, the image compression can add between zero and two additional pixels to the height of a suspended particle.

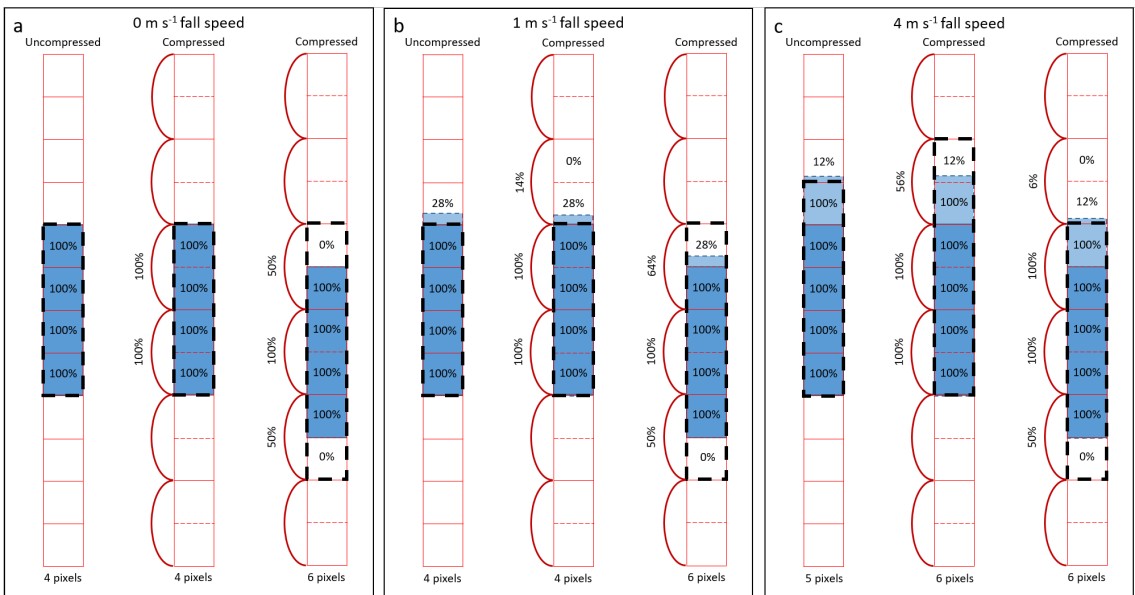

**Figure 3.** Schematic depictions of the combined effects of PIP image compression and motion blurring for a 0.4 mm tall particle (a) suspended in air, (b) falling at 1 m s$^{-1}$, and (c) falling at 4 m s$^{-1}$. The red boxes indicate individual PIP pixels and the dark red braces indicate the pairs of pixels that are averaged together during the image compression. The darker blue shading indicates the area covered by the actual particle at the end of the image exposure period and the lighter blue shading indicates the additional area that appears to be covered by the particle due to motion blurring. The dashed black box indicates the pixels which appear to be contained within the particle (i.e., having $ge$50% coverage), taking into account both motion blur and compression. The value listed inside each pixel indicates the percent coverage of that pixel by the apparent particle prior to compression and the value listed along each brace indicates the percent coverage of the paired pixels post-compression (this value is assigned to both of the paired pixels when the image is decompressed for analysis). In each panel, the left column of pixels represents the view before compression and the right two columns of pixels represent the view after compression, corresponding to the two extremes of how a particle could align to the compression pixel pairs.

For a PIP-viewed particle with a non-zero fall speed, both motion blurring and image compression must be taken into account; schematic representations for particles falling a 1 m s$^{-1}$ and 4 m s$^{-1}$ are depicted in Fig. 3b and Fig. 3c, respectively.

Because of our prescribed threshold of 50% coverage, the 1-m-s$^{-1}$ fall speed case ends up producing the very similar results to the suspended particle case: the addition of between zero and two additional pixels to the height with an average of 1.28 extra pixels at 1 m s$^{-1}$ compared to 1.0 pixels when suspended based on our statistical analysis. For a particle falling at 4 m s$^{-1}$, however, the motion blurring adds an extra pixel to the height prior to compression and the image compression adds an another pixel; in total, the motion blurring and compression effects add between one and two pixels to the height of the particle

(average of 2.12 pixels). Since the image compression only impacts vertical motion blurring, the 10 m s$^{-1}$ particle motion is not included in Fig. 3, but were a snowflake to fall at such an extreme speed, motion blurring would, on average, add 2.80 pixels before accounting for the effects of image compression and 3.80 pixels after accounting for image compression.

Of the three instruments considered in this study, 2DVD has the shortest exposure time at $18\mu$s or $\sim$1/55500th of a second (Schönhuber et al., 2008). Each 2DVD line-scan camera has 512 photodetectors that are calibrated to observe a 100-mm wide field of view (Kruger and Krajewski, 2002), from which we can infer that the pixel size is $\sim$0.195 mm (195 $\mu$m). With the shortest exposure time and largest pixel size, it is unsurprising that 2DVD should be the most resilient to motion blur effects according to our analysis of the camera setups. Even for a snowflake with an extremely fast fall speed of 4 m s$^{-1}$, a single point on the snowflake would only move 72 $\mu$m during the exposure time, which is only $\sim$37% of the pixel size assuming the camera elements gather light isotropically in the vertical and horizontal directions. As with MASC, 2DVD uses an enclosed sampling volume, which reduces the opportunities for particles to have large horizontal motions; however, a particle moving at 10 m s$^{-1}$ would still only add 0.92 pixels on average to the along-motion particle dimension.

The extent to which the motion blurring discussed above might impact the measurement of particle area and, by extension, the equivalent diameter, is highly dependent on the shape and orientation of the particle. The area errors introduced by motion blur will increase as the number of particle pixels located directly below a non-particle pixel (hereafter referred to as top-edge pixels) increases. The motion blur of these top-edge pixels occurs when the particle leaves those pixels during the image exposure period. For a particle with an entirely convex edge, the total number of top-edge pixels will be equal to the horizontal width of the particle, in pixels; for non-convex particles, the number of top-edge pixels will depend on how many additional top-edge pixels, if any, the concave edge regions add to the particle.

Using the statistical analysis detailed above, we determined the average effects of motion blurring (and image compression) on area by multiplying the number of extra particle pixels by the length in pixels of the original particle (i.e., we assume the particles are circular for the area calculations). In the interest of computational simplicity, coverage in the along-motion and across-motion directions are considered independently (i.e., a pixel located such that the across-motion particle width produces 50% coverage of the pixel and the along-motion particle width produces 50% coverage of the pixel would count as a particle pixel even though such a pixel would actually only be 25% covered by the particle). Additionally, we assume that the particle is positioned such that its across-motion extent begins at the edge of a pixel and that the cumulative effects of pixelating a circular particle on apparent pixel coverage are negligible (in actuality, the coverage for each edge pixel would vary across the edge of a circular particle). Finally, we assume that all particles have a non-oblique motion relative to the pixel grid. The results of this analysis are depicted in Fig. 4 with the fractional biases being calculated against the diameter and area that would be observed by each instrument in the absence of motion blurring but with pixelation effects.

While the relative impact of the area and equivalent diameter errors introduced by motion blur depend on the application for which the measurements are being used, we can draw a number of general conclusions from our analysis. Key among these is that very small particles will experience considerably greater relative impacts from the motion blur errors than will larger particles falling at the same rate; this is particularly evident in the fractional bias plots (Fig. 4c,d). Generally speaking, however, smaller particles of a given ice crystal habit will tend to have a slower fall speed than larger particles of that same habit and will, therefore, typically experience less motion blur in the vertical. For a very fast fall speed of 4 m s$^{-1}$, the overestimation of the equivalent diameter for very large circular particles (diameter $\gtrsim$ 10 mm) is approximately two orders of magnitude smaller than the actual equivalent diameter (i.e., a fractional bias of 0.01). Even with a relatively fast horizontal motion of 10 m s$^{-1}$, the

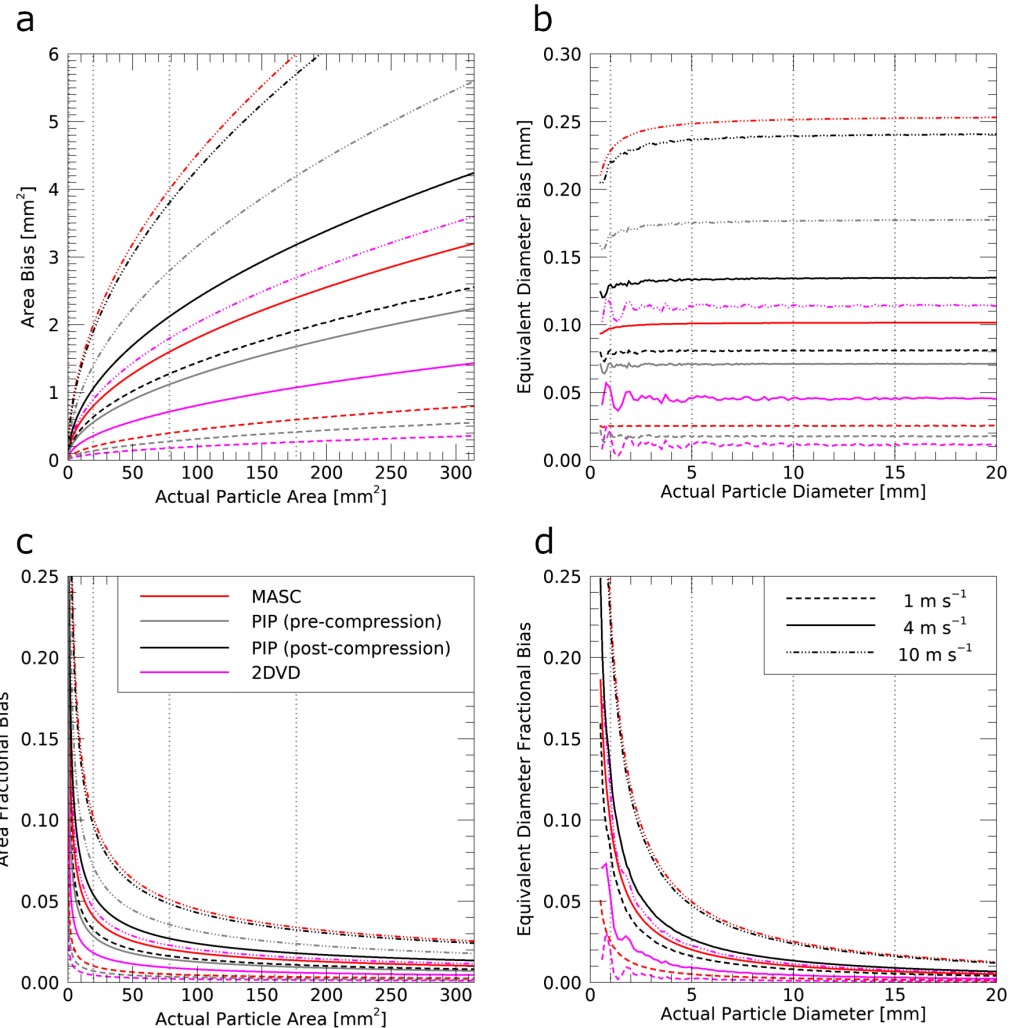

**Figure 4.** Approximate effects of motion blur on (a,c) measured area [mm$^2$] and (b,d) measured equivalent diameter [mm] for hypothetical perfectly circular particles moving at (dashed line) 1 m s$^{-1}$, (solid line) 4 m s$^{-1}$, and (dash-dot line) 10 m s$^{-1}$ as (red) MASC, (gray) PIP before accounting for image compression, (black) PIP after accounting for image compression, and (magenta) 2DVD would measure them. The effects are portrayed both in terms of (a,b) the bias, computed as the difference between the motion-blurred quantity and the actual particle quantity, as well as (c,d) the fractional bias, computed as the bias divided by the non-blurred, non-compressed particle quantity. Gray vertical lines indicate the corresponding quantity for particles with diameters of 1, 5, 10, and 15 mm.

equivalent diameter fractional bias still remains below 0.03 for these very large circular particles. Critical to interpreting and applying these results is the fact that we assumed that the particles are perfectly circular. As the particles under consideration
400  become more oblate, with their horizontal dimension growing relative to their vertical dimension, both the absolute and relative

effects of motion blurring on the measured area and equivalent diameter will increase relative to circular particles of the same area. An oblate particle of equal area to a circular particle will have more top-edge pixels than the circular particle. Increasing the number of top-edge pixels increases the number of pixels that the motion blur is applied to, thereby increasing the absolute effects of the motion blurring. Furthermore, because the particle area is remaining constant, the relative effects of the motion blurring are also increasing. For similar reasons, increasing the complexity of the particle outline such that the number of top-edge pixels increases will also result in larger biases due to motion blur.

Rephrasing this discussion in terms of accuracy, PIP will produce a slightly less accurate measurement of area and equivalent diameter than MASC in the presence of motion blurring, as demonstrated in Fig. 4. Interestingly, were the PIP images not compressed prior to their analysis, the PIP would produce more accurate measurements than MASC. As we established above, the 2DVD is the least affected by motion blur and, as a result, has the most accurate area measurements, although it should be noted that 2DVD also has a relatively large pixel size.

## 5.2 MASC and PIP shape-fitting measurements

As discussed in section 4, the PIP data for the period between 0000 UTC 7 March and 0000 UTC 9 March 2018 at the ICE-POP MHS observation site was processed using algorithms designed to emulate both the MASC and 2DVD measurements of particle dimension. The present section will focus on the measurements made using the MASC and PIP shape-fitting methods as well as the tensor method. A comparison of the bounding box measurement techniques used by PIP and 2DVD will be the subject of a separate section (section 5.3) rather than being included here. Although the tensor-fitted ellipse measurements will be used here as a point of comparison between the various instrument algorithms, it should be noted that the tensor-fitted ellipse measurements are not a 'ground truth' and are subject to errors of their own.

Comparing the PIP-fitted ellipse particle dimensions to those of the tensor-fitted ellipse, we found that the PIP-fitted ellipse tends to overestimate the long dimension of the particle (Fig. 5a), measured as the major axis of the fitted ellipse, and underestimate the short dimension (Fig. 5d), measured as the minor axis of the fitted ellipse, relative to the tensor-fitted ellipse. As a result, the aspect ratio, computed as the short dimension divided by the long dimension, is almost always underestimated by the PIP-fitted ellipse relative to the tensor-fitted ellipse. This underestimation is particularly noticeable for larger tensor-fitted ellipse aspect ratios with the PIP-fitted ellipse aspect ratio rarely exceeding 0.6 and never reaching 0.7 during this event, despite there being a period of lump graupel, which should have a relatively high aspect ratio due to its almost spherical shape, on 8 March (Fig. 6a). This effectively represents an artificial cap in PIP-fitted ellipse aspect ratio.

Using the PIP-fitted rectangle as the basis of the particle dimension measurements produces a sizable improvement to the agreement between the PIP-fitted long dimension and the tensor-fitted long dimension (Fig. 5a,b) but almost no change in agreement for the short dimension (Fig. 5d,e). To quantify these changes in the dimension measurement agreement, we use both the mean absolute difference, defined as the average magnitude of the difference between two sets of measurements, and the mean absolute fractional difference, defined as the average magnitude of the normalized difference between two sets of measurements; for Fig. 5, the normalization is done relative to the measurements made using the tensor method. Comparing the mean absolute differences for the the PIP-fitted ellipse and PIP-fitted rectangle dimensions, the PIP-fitted rectangle long

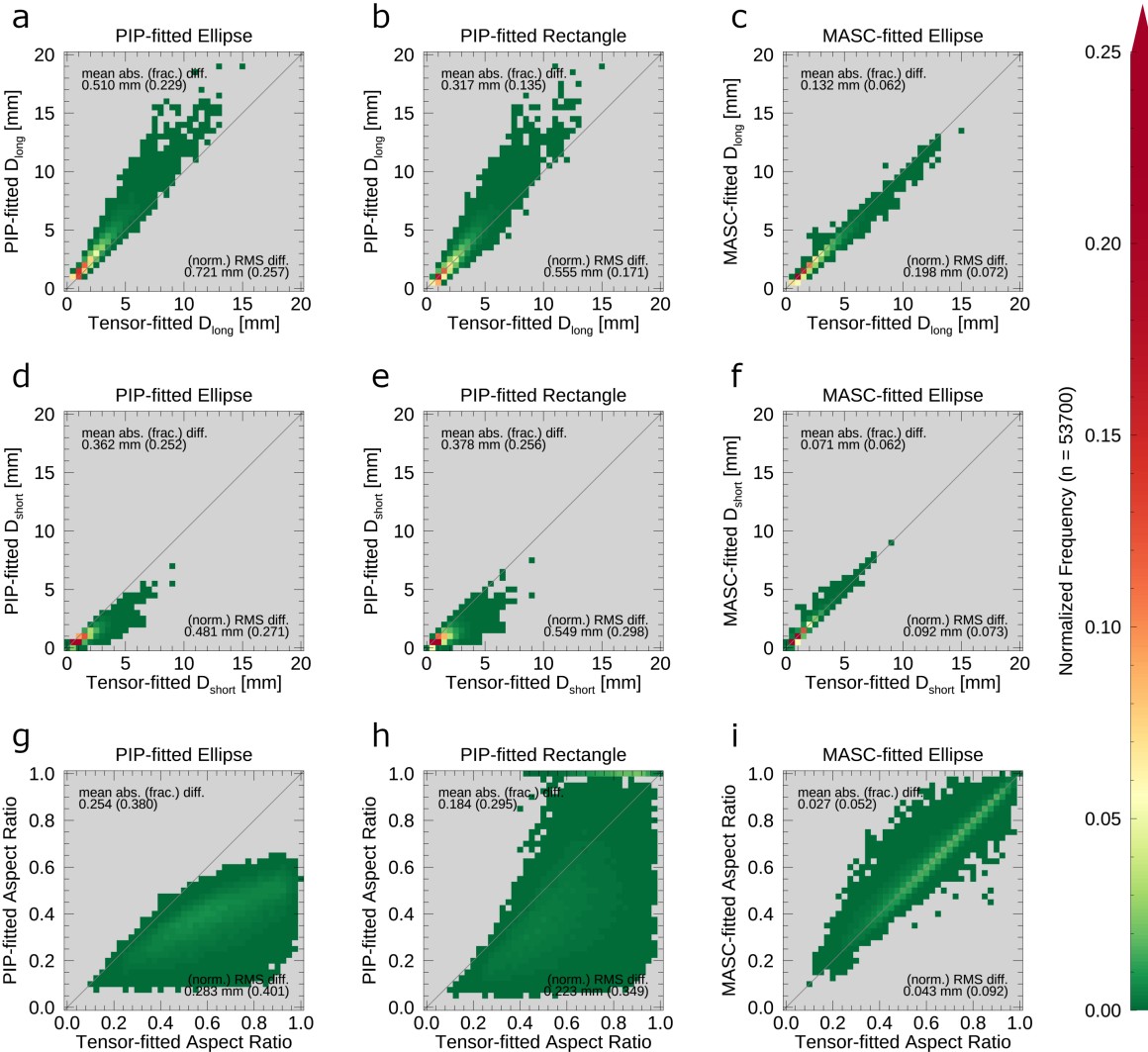

**Figure 5.** Two-dimensional histograms of (a-c) particle long dimension in mm, (d-f) particle short dimension in mm, and (g-i) particle aspect ratio for (a,d,g) the PIP-fitted ellipse, (b,e,h) the PIP-fitted rectangle, and (c,f,i) the emulated MASC-fitted ellipse as a function of the tensor-fitted ellipse values. The diagonal grey lines indicate where the x value is equal to the y value. The mean absolute difference and root-mean-square difference for each pair of measurements is indicated on each panel with the mean absolute fractional difference and normalized root-mean-square difference in parentheses. Only particles with a PIP-measured area greater than 0.5 mm$^2$ are included. Note, a logarithmic color table has been used to highlight the frequency distribution at lower values.

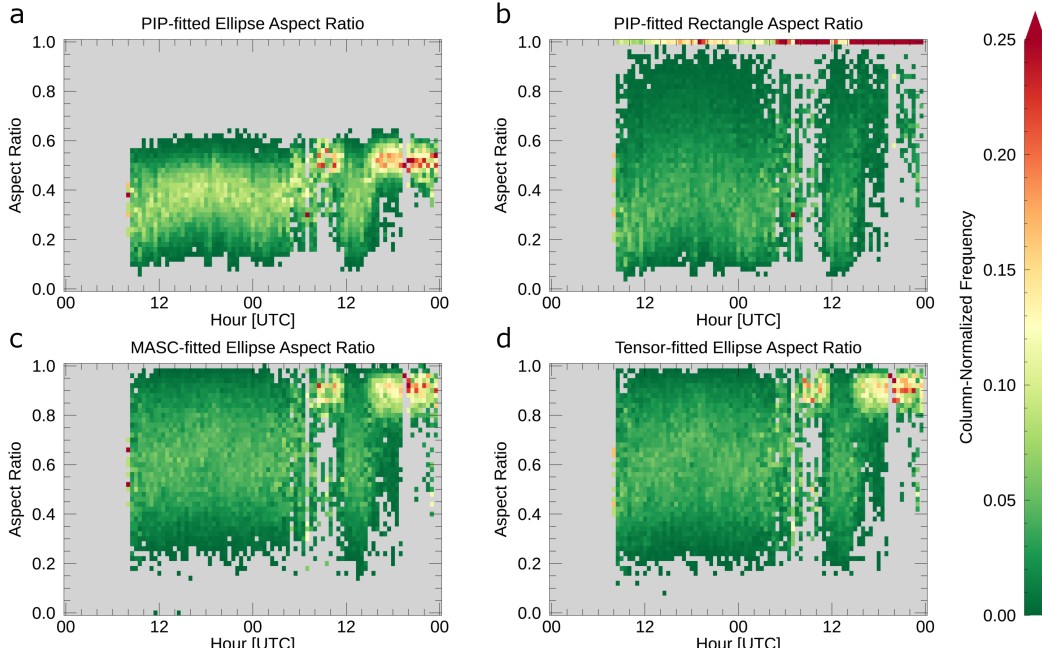

**Figure 6.** Time series of the (a) PIP-fitted ellipse and (b) PIP-fitted rectangle, (c) emulated MASC-fitted ellipse, and (d) tensor-fitted ellipse aspect ratio distributions for the 7–8 March 2018 case. Only particles with an area of at least 0.5 mm$^2$ were included in the histograms. The bin counts of each column have been normalized such that they add up to one.

dimension has a mean absolute difference from the tensor-fitted long dimension that is approximately three-fifths that of the PIP-fitted ellipse long dimension (Fig. 5a,b). The PIP-fitted ellipse and rectangle short dimensions, however, show almost no change in mean absolute difference (Fig. 5d,e).

In terms of aspect ratios, the PIP-fitted rectangle aspect ratio (Fig. 5h) does not suffer from the artificial cap that was present with the PIP-fitted ellipse aspect ratio (Fig. 5g) and much of the reduction in the mean absolute difference (0.254 versus 0.184)
is likely tied to the lack of said artificial cap. That said, the PIP-fitted rectangle aspect ratio still has a tendency to greatly underestimate the aspect ratio relative to the tensor-fitted ellipse aspect ratio (Figs. 5h and 6b,d). In contrast to the PIP-fitted ellipse aspect ratios, the PIP-fitted rectangle aspect ratio does capture the large aspect ratios associated with the periods of lump graupel precipitation on 8 March around 0900 UTC and after 1400 UTC, although these aspect ratios are almost entirely reported as a value of one. Interestingly, the PIP-fitted rectangular aspect ratio frequently has a value of one but very rarely has
a value between 0.9 and one; this peculiarity will be revisited later in this section.

In contrast to the two sets of PIP-fitted measurements, the MASC-fitted ellipse measurements show very good agreement with the tensor-fitted ellipse measurements of both the long and short dimension of the fitted particles (Fig. 5c,f) with mean absolute differences ranging between approximately one-fifth and two-fifths those for the PIP-fitted shapes. As a result, the aspect ratios computed from shapes fitted using the emulated MASC method and the tensor method are in fairly good agreement

as well (Fig. 5i). This general agreement between the particle dimensions measured from the MASC-fitted and tensor-fitted ellipses suggests that these two shape fitting algorithms may give more reliable results than the PIP-fitted ellipse or rectangle algorithms. To gain insight into whether the MASC and tensor method algorithms are, in fact, performing better than the PIP algorithms, we will take a closer look at individual particles.

Figure 7a-d depicts snapshots of particles taken from the PIP AVI files with the fitted shapes from the various fitting algo-
rithms depicted in Fig. 7e-h. For simplicity, these particles will be referred to by the Fig. 7 panel letter of the unannotated panels (i.e. panels a–d). Particles (a) and (b) are both likely some type of aggregate frozen precipitation based on their odd shapes. Based on the relatively circular shapes of the remaining two particles, relatively high fall speeds (black line, Fig. 1), subfreezing near-surface temperatures (red line, Fig. 1), and the lack of an above freezing temperature layer in a nearby thermodynamic sounding (not shown), particles (c) and (d) are likely both examples of lump graupel. Qualitative and quantitative inspection
of the particles in Fig. 7 and Table 1 supports the patterns identified by the scatter-plot analysis performed above. Specifically, the PIP-fitted ellipse appears to always overestimate the particle long dimension; the PIP-fitted rectangle appears to some-times overestimate, sometimes underestimate, and sometimes accurately capture the particle long dimension; and, finally, the emulated MASC-fitted and the tensor-fitted ellipses tend to produce fairly similar particle dimensions. Additionally, visual examination of these and other (not shown) individual particles suggests that the emulated MASC-fitted and the tensor-fitted
ellipses tend to provide more reasonable estimates of particle dimension than either of the PIP-fitted shapes.

The primary cause of the discrepancies between the actual particle shape and the PIP-fitted ellipse major and minor axes is rooted in the reliance on only the particle area and perimeter for making the PIP fits, as discussed in section 3.1. For particles with complicated outlines, such as particle (b), the particle perimeter is far greater than the perimeter of an ellipse or rectangle of equal dimensions (i.e., length and width). To make up the extra perimeter for a given area, the PIP-fitted ellipse or rectangle
must have a larger long dimension at the expense of the short dimension.

The relationship between excess perimeter for a given area, relative to a perfect ellipse, and error in the PIP-fitted ellipse or rectangle dimension lengths is demonstrated in Fig. 8 for a hypothetical particle that we will assume is generally circular with a mean radius of 0.5 mm, but whose edge is not a perfect circle and is allowed to vary about the mean radius of 0.5 mm in such a way that the area and center position remain unchanged from that of a 0.5-mm-radius circle. The amplitude of this variation
in radius is represented in Fig. 8 by the perimeter stretching factor, computed as the actual perimeter of a particle divided by the perimeter of a circle of equal area to the particle. A perimeter stretching factor of one will result in a perfect circle while a larger perimeter stretching factor indicates the presence of larger variations in radius (i.e., a more complicated shape). For our hypothetical particle, we prescribe a value between one and two for the perimeter stretching factor and set the particle area to be equal to that of a 0.5-mm-radius circle. As the PIP-fitting method only requires perimeter and area, our parameterization
of the hypothetical particle is sufficient to perform an ellipse or rectangle fit without constructing the hypothetical particle itself. The equations used to make the PIP fits are derived in appendix A. Note that nothing in these equations relates to the actual shape of the particle itself (e.g., circular or elliptical); this missing information is, in fact, the core issue with both the PIP ellipse and rectangle fits. Without knowledge of the underlying distribution of pixels within the particle, the PIP algorithm assumes all particles are either smooth-edged ellipses or rectangles, depending on which fit is used.

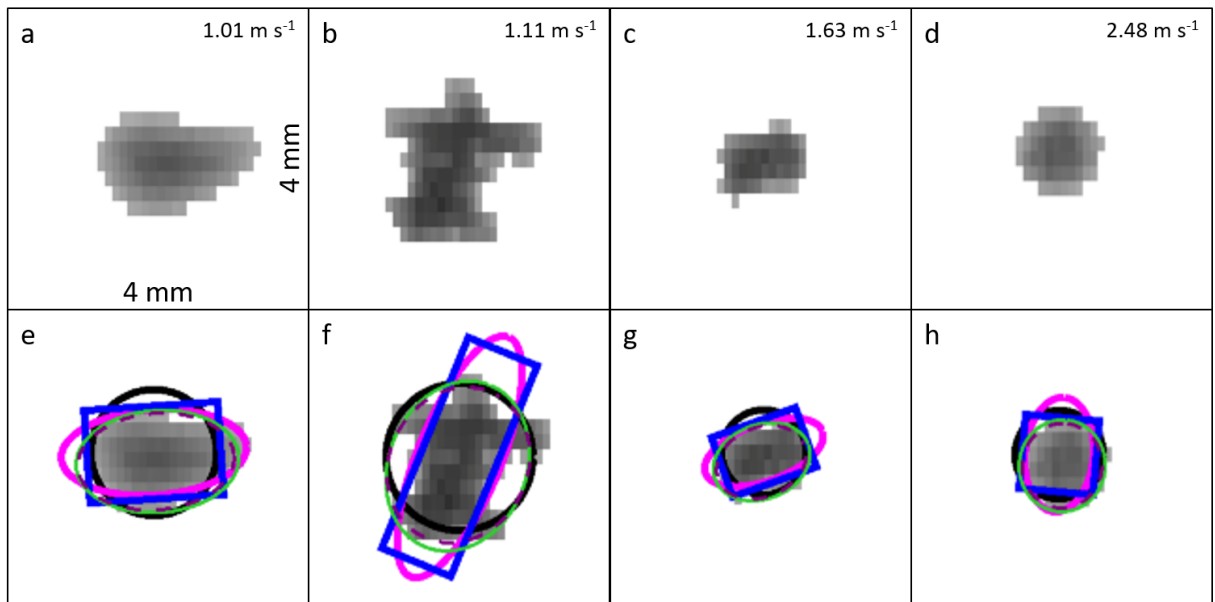

**Figure 7.** Snapshots of four particles as viewed by PIP collected by PIP at approximately (a,b,e,f) 1904 UTC 7 March 2018 and (c,d,g,h) 1756 UTC 8 March 2018 with (e-h) overlaid particle shape estimates as determined by the (black) area-equivalent circle, (magenta) PIP-fitted ellipse, (blue) PIP-fitted rectangle, (purple dashed) emulated MASC-fitted ellipse, and (green) tensor-fitted ellipse. Each panel represents a square that is 4-mm long on each side. The dimensions of the fitted shapes for each particle are presented in Table 1. Fall speeds are listed in the top right corner of (a-d); the fall speeds for particles (a) and (b) are computed by PIP while the fall speeds for particles (c) and (d) are manually calculated as the average frame-to-frame center position motion as these specific particles were not included in the PIP particle velocity files.

The lack of shape information in the PIP shape-fitting equations does not entirely explain the artificial cap in aspect ratio at ∼0.6 for the PIP-fitted ellipses (as in Figs. 5g and 6a), however. If the shape issues were the sole cause of the artificial cap, the circular presentation of particle (d) in Fig. 7 would produce a very good fit with an aspect ratio approaching one. Instead, the aspect ratio cap is a result of where the aspect ratio is most sensitive to the perimeter length for a given particle area. Based on Fig. 8a, the PIP-fitted ellipse major and minor axis lengths and aspect ratio show the greatest sensitivity to

perimeter length when the perimeter stretching factor is almost one (i.e., the particle is almost a perfect circle). As a result, any deviation from the smooth edge of a perfect circle will produce relatively rapid changes in the ellipse axis lengths and the aspect ratio. It was noted above that the PIP-fitted ellipse measurements (Fig. 6a) appear to produce an artificial cap on aspect ratio of approximately 0.6. Based on our calculations for Fig. 8, a PIP-fitted ellipse aspect ratio of 0.6 is produced when the perimeter stretching factor is approximately 1.065 regardless of the actual shape of the particle. For reference, a perimeter

stretching factor of 1.065 corresponds to taking a 10-mm diameter circular particle, which would have a perimeter of ∼31.4 mm, and stretching its perimeter by 2 mm. Small increases in perimeter, such as this, can be introduced by a few very small

**Table 1.** Dimensions and aspect ratios of the fitted shapes for various particle fits depicted in Fig. 7, referenced by the figure panel letter for the unannotated particle images. The perimeter stretching factor is computed by dividing the actual particle perimeter by the perimeter of a circle of equal area to the particle.

| Fitted-shape measurement | Fig. 7 panel | | | |
| --- | --- | --- | --- | --- |
| | (a) | (b) | (c) | (d) |
| Equivalent diameter [mm] | 1.680 | 1.950 | 1.130 | 1.180 |
| Perimeter stretching factor | 1.148 | 1.308 | 1.144 | 1.068 |
| **PIP-fitted ellipse** | | | | |
| Major-axis length [mm] | 2.478 | 3.434 | 1.660 | 1.529 |
| Minor-axis length [mm] | 1.141 | 1.105 | 0.767 | 0.916 |
| Aspect ratio | 0.460 | 0.322 | 0.462 | 0.599 |
| **PIP-fitted rectangle** | | | | |
| Long-side length [mm] | 1.790 | 3.020 | 1.190 | 0.990 |
| Short-axis length [mm] | 1.240 | 0.990 | 0.840 | 0.990 |
| Aspect ratio | 0.693 | 0.328 | 0.706 | 1.000 |
| **Emulated MASC-fitted ellipse** | | | | |
| Major-axis length [mm] | 2.093 | 2.121 | 1.238 | 1.132 |
| Minor-axis length [mm] | 1.288 | 1.831 | 0.926 | 1.065 |
| Aspect ratio | 0.615 | 0.863 | 0.748 | 0.941 |
| **Tensor-fitted ellipse** | | | | |
| Major-axis length [mm] | 2.178 | 2.311 | 1.331 | 1.227 |
| Minor-axis length [mm] | 1.363 | 1.831 | 0.990 | 1.150 |
| Aspect ratio | 0.626 | 0.792 | 0.744 | 0.938 |

deviations of the particle edge from a perfect circle as well as by the inability to perfectly represent a circle using square pixels (i.e., pixelation effects).

The PIP-fitted rectangle long- and short-side lengths (Fig. 8b) have a similar sensitivity to changes in perimeter stretching factor when the perimeter stretching factor is relatively small. A key difference from the PIP-fitted ellipses, however, is that a rectangle cannot be fit to a particle whose perimeter stretching factor is less than $\sqrt{4/\pi}$ ($\sim$1.128), below which the particle perimeter is not long enough to construct a rectangle of the required area. Particle (d) in Fig. 7 is an example of a particle whose actual perimeter is too short to produce a physical rectangle fit using the PIP method. The perimeter stretching factor for particle (d) is approximately 1.068, as listed in Table 1. Based on this and similar particles, the PIP processing algorithm appears to automatically assign an aspect ratio of one to any particle where a rectangle fit is not physically possible, the effects of which can be seen in Fig. 6b. How the PIP algorithm determines a dimension length in these cases, however, remains unclear.

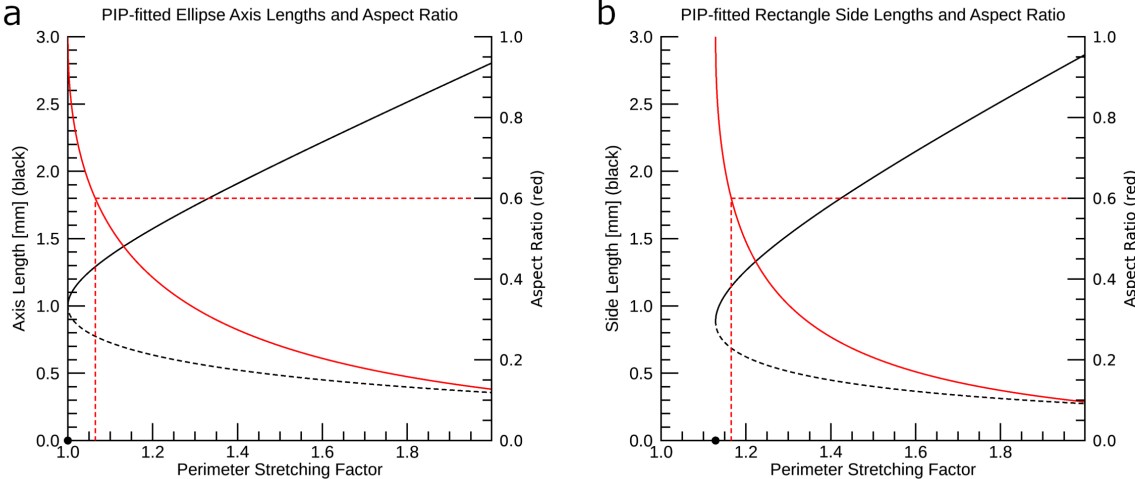

**Figure 8.** Demonstration of the impact of excess perimeter for a given area on the (a) PIP-fitted ellipse and (b) PIP-fitted rectangle long-axis length (black solid), short-axis length (black dashed), and aspect ratio (red solid). This example uses a circle with a radius of 0.5 mm to compute the base perimeter and area. The red dashed lines highlight the perimeter stretching factor that corresponds to an aspect ratio of 0.6 and the black dot indicates the minimum perimeter stretching factor for which a physically meaningful shape can be fitted.

The artificial cap in aspect ratio that appears for the PIP-fitted ellipses (Fig. 6a) does not appear to be present in the aspect ratio distributions for the PIP-fitted rectangles (Fig. 6b). A key factor that leads to this difference is demonstrated in Fig. 8b: a rectangle aspect ratio of 0.6 corresponds to a perimeter stretching factor of ∼1.165 compared to 1.065 for a PIP-fitted ellipse (Fig. 8a). The importance of this perimeter stretching factor difference of 0.1 comes down to the observed distribution of particle perimeters relative to their areas (i.e., the distribution of particle stretching factors). Figure 9 depicts the distribution of perimeter stretching factors for the 7–8 March 2018 case. Only a very small portion of particles have a perimeter stretching factor smaller than 1.065 (left red dashed line, Fig. 9) whereas the most frequently observed perimeter stretching factors are smaller than the 1.165 perimeter stretching factor (right red dashed line, Fig. 9) that corresponds to the 0.6 aspect ratio for a PIP-fitted rectangle.

### 5.3 2DVD and PIP bounding-box measurements

Because the 2DVD does not instantaneously capture full images of a particle, a 'long dimension versus short dimension' aspect ratio is not computed by the instrument. Instead, 2DVD reports the 'oblateness' of a particle, defined as the aspect ratio of the horizontal width of a particle versus its vertical height. For each camera, the oblateness is computed as the bounding-box height divided by the bounding-box width. An overall oblateness is then computed from the two single-camera oblateness values by their geometric mean. Here we will focus on the measurements of the bounding box height and width rather than the oblateness itself.

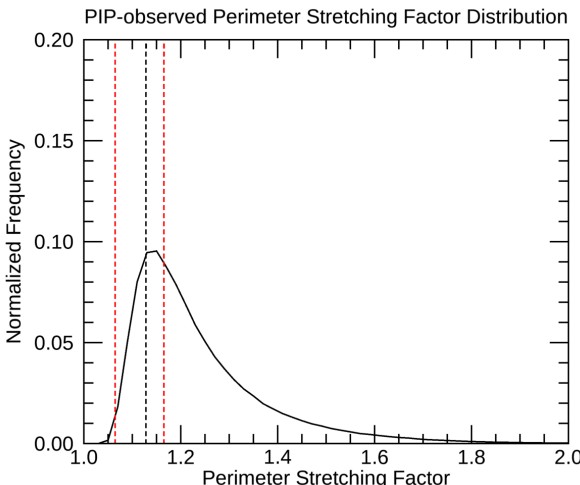

**Figure 9.** Distribution of perimeter stretching factors computed from PIP observations collected during the 7–8 March 2018 snow event. The two red vertical dashed lines indicate the perimeter stretching factor corresponding to an aspect ratio of 0.6 for a (left red dashed line) PIP-fitted ellipse and (right red dashed line) PIP-fitted rectangle. The black vertical dashed line indicates the smallest perimeter stretching factor that will permit a physically meaningful rectangle fit. Only particles with an area of at least 0.5 mm$^2$ are included in the histogram.

Both PIP and 2DVD record measurements of the horizontal and vertical extent of a particle in the form of bounding-box measurements, computed as the horizontal distance between the left-most and right-most particle pixels and the vertical dis-
tance between the bottom-most and top-most particle pixels. Because PIP takes measurements using two-dimensional images, the accuracy of the PIP bounding-box measurements is directly tied to the resolution of the camera and the degree of motion blurring. 2DVD, however, uses a line-scan camera and, as such, the 2DVD bounding-box measurements are derived quantities. The bounding-box height is derived from the time required for the particle to pass through the sampling region while the bounding-box width is computed by compositing the individual line scans to rebuild the particle. As discussed in section
3.3, 2DVD uses an unskewing algorithm to correct the bounding-box width measurements for horizontal particle motion. As the actual 2DVD unskewing algorithm is part of a proprietary software package, we use a conceptually similar unskewing algorithm, described in section 4.2.

The vertical sampling frequency of the emulated 2DVD images is usually much higher than that of the underlying PIP images due to the high temporal sampling frequency of 2DVD. In order for the vertical sampling frequency of the emulated
2DVD images to be lower than that of the underlying PIP images, the precipitation particle would need to have a fall speed greater than $\sim$68.2 m s$^{-1}$, which is unrealistic for any of the particles of interest. Because of this, we would expect the emulated 2DVD bounding-box height to almost exactly match the PIP-measured bounding-box height and this expectation is borne out: the mean absolute difference between the PIP and 2DVD bounding-box heights is only 0.016 mm (a mean absolute fractional difference of 0.013).

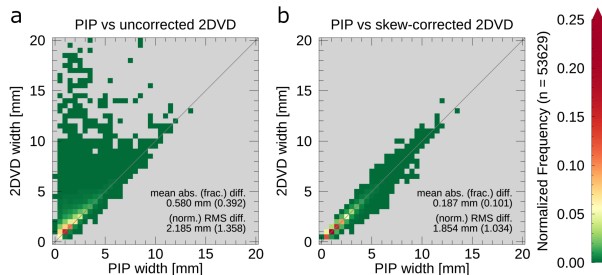

**Figure 10.** Two-dimensional histograms comparing the PIP bounding-box width to the (a) uncorrected and (b) skew-corrected 2DVD bounding-box width. The mean absolute difference and root-mean-square difference for each pair of measurements is indicated on each panel with the mean absolute fractional difference and normalized root-mean-square difference in parentheses. Only particles with a PIP-measured area of at least 0.5 mm$^2$ are included in the plots. Note, a logarithmic color table has been used to highlight the frequency distribution at lower values.

Of greater interest is how well the emulated 2DVD bounding-box width measurements perform, especially after we apply our approximation of the 2DVD unskewing algorithm. Figure 10a compares the uncorrected emulated 2DVD bounding-box width with the PIP-measured bounding-box width. Unsurprisingly, the uncorrected 2DVD measurements tend to greatly overestimate the width of the bounding box due to the skewing introduced by any horizontal motion of the particle. One point of interest, however, is that the uncorrected 2DVD measurements can sometimes underestimate the bounding box width. This can occur

when a particle of sufficiently low aspect ratio is moving in the opposite direction of the tilt of the top of the particle (e.g., a needle crystal whose top is to the left of the particle centroid moving towards the right); this will result in the particle being compressed in the horizontal because the particle is scanned from the bottom upwards as it falls through the plane of the line scan camera.

The emulated 2DVD measurements of bounding-box width are surprisingly accurate once our approximation of the 2DVD

unskewing algorithm is applied (Fig. 10b). Quantitatively, the unskewing algorithm produces a reduction in mean absolute difference from 0.58 mm to 0.187 mm. After the correction, the overestimation of bounding-box width is considerably reduced compared to the uncorrected emulated 2DVD measurements. The underestimation of bounding-box width that was present with the uncorrected measurements, however, is more common after the correction. The increase in underestimation is due to the assumption that the top-most and bottom-most pixels of the 2DVD-imaged particle are supposed to be vertically aligned with

one another. This assumption is frequently broken by snow particles, leaving the 2DVD measurements of snowflake horizontal extend potentially unreliable. An example of this assumption resulting in a very minor underestimation of bounding-box width can be seen in Fig. 2.

## 6 Discussion

Previous studies have examined a number of error sources that impact the accuracy of optical disdrometer measurements. To better understand the magnitude of the differences examined in the present study, it is worth a brief comparison to the findings of past studies.

Simulating two-dimensional observations of three-dimensional particles for the SVI (the predecessor of PIP) and 2DVD, Wood et al. (2013) found that, under reasonable conditions, using a two-dimensional projected long dimension length would result in the observed long dimension being approximately 82% of the actual particle long dimension, thereby underestimating reflectivity by 3.2 dB. Using three-dimensional snowflake replicas, Leinonen et al. (2021) determined that the two-dimensional measurements of particle long dimension made by MASC have a normalized root-mean-square error of ∼6% relative to the actual particle long dimension. Leinonen et al. (2021) only examined measurements of snowflakes with a long dimension between 3 mm and 5 mm and the study used the largest long dimension measured among the three MASC cameras rather than using a single-camera measurement. By comparison, we found the normalized root mean square difference between the long dimension of the tensor-fitted ellipse and the PIP-fitted ellipse, PIP-fitted rectangle, and MASC-fitted ellipse to be ∼26%, ∼17%, and ∼7%, respectively, and these numbers do not change much when only particles with a tensor-fitted ellipse long dimension between 3 mm and 5 mm are included in the calculation. This suggests that the PIP shape-fitting issues have a much greater impact on the determination of the particle long dimension than does the loss of information caused by projecting a three-dimensional snowflake into a two-dimensional view. Interestingly, it also suggests that differences between the MASC-fitted ellipse and tensor-fitted ellipse are slightly larger than the error introduced by using the two-dimensional projection of the particle to measure long dimension length, although this could be due to Leinonen et al. (2021) using the maximum long dimension from among the three MASC cameras rather than just using a single camera.

Fitch et al. (2021) found that the largest snowflakes are less likely to be captured by MASC in strong winds ($> 5$ m s$^{-1}$) than other snowflakes. This bias towards collecting smaller particles was especially prevalent when the MASC did not have any form of wind shielding. For the original tall-form 2DVD, however, Nešpor et al. (2000) found that small raindrops were more likely to be caught in the turbulence produced by the wind passing over the instrument and that, under certain conditions, this could prevent these small raindrops from reaching the sampling volume. It is not clear from their study, however, whether this sampling bias would extend to frozen precipitation nor the extent to which the newer low-profile 2DVD addresses this issue. Newman et al. (2009) demonstrated that the SVI instrument setup produces negligible wind shielding effects at the sampling volume. As the PIP setup is virtually identical to the SVI setup, this finding should also apply to PIP. As detailed in the previous section, the present study examined the impact of wind on the snowflake measurements in terms of motion blurring. While we have not specifically examined the expected horizontal particle motions within the sampling volumes of each instrument, it is expected that the open sampling volume of the PIP would make it more prone to horizontal motion blurring than would the closed sampling volumes of 2DVD and MASC. Recent work using observations from a PIP located in Haukelister, Norway, demonstrated reliable results during moderate to high wind conditions for multiple snowfall events over a winter season (Shates et al., 2021). Additionally, these PIP observations were used in conjunction with a MASC to retrieve liquid water equivalent

snowfall rates that corresponded well with accumulation observations from a collocated double fence automated reference standard (Schirle et al., 2019; Wolff et al., 2015). The findings of these studies suggest that the greater potential for horizontal motion blurring of an open sampling volume compared to a closed sampling volume may be relatively minor, at least for snow accumulation applications.

## 7   Conclusions

The present study set out to evaluate the measurement techniques used by PIP, MASC, and 2DVD to characterize the size and shape of snowflakes in terms of area, equivalent diameter, and aspect ratio. This was accomplished by using imagery collected by a PIP instrument during the ICE-POP 2018 field campaign to emulate the relevant measurements made by MASC and 2DVD. More specifically, a comparison was made between the ellipse- and rectangle-fitting algorithms used by PIP and the ellipse-fitting algorithm used by MASC; a tensor-based ellipse-fitting algorithm was also evaluated alongside the PIP and MASC shape-fitting algorithms. Additionally, measurements of the bounding-box width and height made by PIP were compared to the emulated 2DVD width and height measurement in order to evaluate the performance of 2DVD when measuring snow particles. The key findings of our comparative evaluation are listed below.

- Even when exposed to extreme motion blurring ($10 \text{ m s}^{-1}$ particle motion), all instruments had an area bias of less than 10% and an equivalent diameter bias of less than 5% for particles with an actual equivalent diameter larger than 5 mm (section 5.1).

- The 2DVD camera setup provides the most accurate measure of area (and equi-area diameter) due to its resilience to motion blurring, although it should be noted that 2DVD also has a relatively large pixel size (section 5.1).

- The accuracy of PIP area and equivalent diameter measurements is dependant on the direction of motion blurring due to the vertical image compression used by PIP; PIP is more resilient to horizontal motion blurring and less resilient to vertical motion blurring in comparison to MASC (section 5.1).

- The MASC ellipse-fitting algorithm produces reasonable ellipse fits of the particles and, as such, produces reliable aspect ratios (section 5.2).

- The PIP shape-fitting algorithms do not perform well due to their reliance on only the area and perimeter of a particle leading to a tendency towards overestimating the long dimension and underestimating the short dimension that is highly sensitive to small deviations from the smooth-edged fitted shape; this renders the following variables unreliable: 'Ellip_Maj', 'Ellip_Min', 'Rec_BS', and 'Rec_SS' (section 5.2); other variables reported by the PIP are unaffected by the shape-fitting issues.

- The tensor method performed comparably to the MASC ellipse-fitting algorithm (section 5.2).

- Our implementation of the 2DVD unskewing algorithm, designed to remove the effects of horizontal motion from the 2DVD bounding-box-width measurements, performed surprisingly well with snow particles considering the instrument

was designed to measure rain drops. That said, the corrected bounding-box-width measurements are still prone to error due to the motion skewing effects. We expect that the actual, proprietary implementation of this algorithm would have a similar level of performance (section 5.3).

625

These results suggest that all three instruments have applications for studying snow, albeit with differing degrees of effectiveness. The MASC and PIP cameras, which capture two-dimensional images directly, are better suited for measuring snowflake shape than the line-scan cameras used by 2DVD as the MASC and PIP cameras are not susceptible to the image skewing produced by the horizontal motion of a particle that is experienced by 2DVD. That said, the very short exposure time of the 2DVD camera results in high resilience to motion blur, making any measurements of two-dimensional projected particle area highly accurate. PIP suffers from the issue that the shape-fitting routines do not perform well on frozen precipitation particles and, as a result, the PIP-fitted ellipse and rectangle dimension (and, therefore, aspect ratio) measurements are unreliable. Worth noting, however, is that the PIP shape-fitting issues only impact the four variables listed in the relevant bullet point above; the other PIP reported measures of particle size (e.g., particle bounding box dimensions, area, equivalent diameter) remain reliable.

635 As such, any products that use a non-shape-fitting measure of particle size (e.g., Pettersen et al., 2020) remain unaffected by the shape-fitting issue. As the present study has demonstrated, the PIP imagery can be reprocessed and reliable measurements of the long dimension and aspect ratio can be made via the application of an alternative ellipse-fitting algorithm, such as the MASC or tensor-based algorithms. While not demonstrated here, it may be possible to also implement a shape-fitting algorithm for 2DVD using the reconstructed images captured by the line-scan camera, although the reliability of the resulting shape

640 measurements from such an algorithm would need further investigation to test the impacts of the image skewing.

*Data availability.* Bliven, Larry. 2020. GPM Ground Validation Precipitation Imaging Package (PIP) ICE POP. Dataset available online from the NASA Global Hydrology Resource Center DAAC, Huntsville, Alabama, U.S.A. DOI: http://dx.doi.org/10.5067/GPMGV/ICEPOP/PIP/DATA101

## Appendix A: Derivation of shape-fitting equations for PIP

According to the IMAQ Vision concepts manual (National Instruments, 2000), the equations to fit either an ellipse or a rectangle

645 can be derived from the equations for area and perimeter. The area and perimeter equations for a rectangle are, respectively,

$$A = ab \quad \text{and} \tag{A1}$$

$$P = 2(a+b), \tag{A2}$$

where $A$ is the actual particle area, $P$ is the actual particle perimeter, and $a$ and $b$ are the lengths of the long and short sides of

650 the fitted rectangle. For an ellipse, PIP uses the following equations for area and perimeter, respectively:

$$A = \pi ab \quad \text{and} \tag{A3}$$

$$P = \pi \sqrt{2\left(a^2 + b^2\right)}, \tag{A4}$$

where the major axis length is equal to $2a$ and the minor axis length is equal to $2b$. Note, the equation PIP uses for the perimeter of an ellipse is actually an upper bound on the perimeter of an ellipse rather than an actual estimate of the perimeter (Jameson, 2014).

The derivation for the equations used to compute the dimensions of both the PIP-fitted ellipse and rectangle proceed similarly: solving the area equation for either $a$ or $b$ and substituting the result into the perimeter equation. This substitution is followed by putting the resulting equation into the standard form for a quadratic equation for the rectangle and a quartic equation for the ellipse. The standard form quadratic equation for the rectangle fit is

$$0 = -x^2 + \frac{P}{2}x - A, \tag{A5}$$

where $x$ is either $a$ or $b$ depending on the variable for which the area equation was solved. Similarly, the standard form quartic equation for the ellipse fit is

$$0 = -\pi^2 x^4 + \frac{P^2}{2}x^2 - A^2. \tag{A6}$$

Note that the IMAQ Vision concepts manual incorrectly records the standard form equation for the ellipse fit as being identical to that of the rectangle fit.

The standard form equations for both the rectangle- and ellipse-fits can then be solved for their roots. As the ellipse-fit equation is a biquadratic special case of a quartic equation, it can be solved using the same method as a quadratic equation by substituting $z = x^2$, determining the roots, and then substituting $x = \pm\sqrt{z}$ into those roots. Solving both equations via the these methods gives roots of

$$x = \frac{P}{4} \pm \frac{\sqrt{\frac{P^2}{4} - 4A}}{2} \tag{A7}$$

for the PIP-fitted rectangle and

$$x = \pm\sqrt{\frac{P^2}{4\pi^2} \pm \frac{\sqrt{\frac{P^4}{4} - 4\pi^2 A^2}}{2\pi^2}} \tag{A8}$$

for the PIP-fitted ellipse. The only physically meaningful roots will be the positive roots. In the case of the quartic ellipse equation, one of the positive roots will give the value of $a$ and the other will give the value of $b$. The other dimension for the rectangle fit can be computed by plugging the positive root into the equation for area and solving for the unknown dimension length (either $a$ or $b$).

**Appendix B: PIP video file processing**

The following builds on section 4 by further discussing the steps taken to reprocess the PIP video files in order to produce the emulated MASC and tensor method fits as well as the emulated 2DVD measurements. Reprocessing the PIP video files for the

**Table B1.** List of relevant variables within PIP data files for the reprocessing PIP AVI video files.

| Variable | File | Description |
|---|---|---|
| recnum | _a_p.dat, _a_v_1.dat, _a_v_2.dat | Record number of particle measurement, unique within each 10-minute period |
| Q, R | _a_p.dat | Frame index split into two numbers, unique within each 10-minute period. |
| x_cent, y_cent | _a_p.dat | PIP-measured particle centroid location in pixels |
| vel_v_1, vel_v_2 | _a_v_1.dat, _a_v_2.dat | PIP-measured particle vertical fall speed in m s$^{-1}$ |

alternative fits requires four sets of files: the AVI video files, the '_a_p.dat' particle table files, the '_a_v_1.dat' velocity table files, and the '_a_v_2.dat' velocity table files. The two velocity table files cover different sets of particles: the '_a_v_1.dat' files contain data for particles that were only observed in two consecutive frames while the '_a_v_2.dat' files contain data for all particles that were observed in three or more consecutive frames. Using both files maximizes the number of particles that we can use to make emulated 2DVD measurements. Each AVI file corresponds to one of each of the '.dat' files, although the '.dat' files typically cover a much greater period of time than what is depicted in the AVI files. The key '.dat' file variables for the reprocessing algorithm are listed in Table B1.

The PIP AVI videos include both the PIP view as well as a header that indicates which PIP unit recorded the observations, the time stamp, a combined 'Q_R' frame index, and the number of the frame within the AVI video file (Fig. B1). The key to matching a single frame of the AVI file to the data contained within the '.dat' files is matching the single-number 'Q_R' frame index depicted in the video to the two-number 'Q' and 'R' frame index listed in the '_a_p.dat' files; 'Q_R' can be computed as 'Q' times 32767 plus 'R'. To obtain the 'Q_R' frame index from the video image, we constructed a simple image matching algorithm. The image matching algorithm compares, via subtraction, the image of each digit of the 'Q_R' frame index to a set of images of individual digits that were previously cropped from PIP AVI videos. A matched digit should result in a difference of zero, although we selected the digit that produces the smallest difference and allowed for a difference of up to 100, noting that the 'Q_R' frame index can be between one and six digits long and each additional digit shifts the position of all the numbers in the frame by several pixels.

Once the 'Q_R' frame index is determined from the video file and it has been matched to the 'Q' and 'R' frame index in the '_a_p.dat' file, the record number ('recnum') in the '_a_p.dat' file can be matched to the record number in the '_a_v_1.dat' and '_a_v_2.dat' velocity table files. Note, only a subset of particles listed in the '_a_p.dat' file will have matching entries in the velocity table files. Following the above method, we should now have the centroid location and fall speed of each particle in a given frame of the PIP AVI video file. The centroid is used to identify all pixels within a particle, as described in section 4, and the fall speed is used to emulate the 2DVD measurements, as described in section 4.2.

*Author contributions.* Dr. Helms led this research by developing and running the code to reprocess the PIP video files, implementing the various measurement algorithms, and performing the various analyses. Dr. Munchak and Dr. Tokay provided input on the methods used in

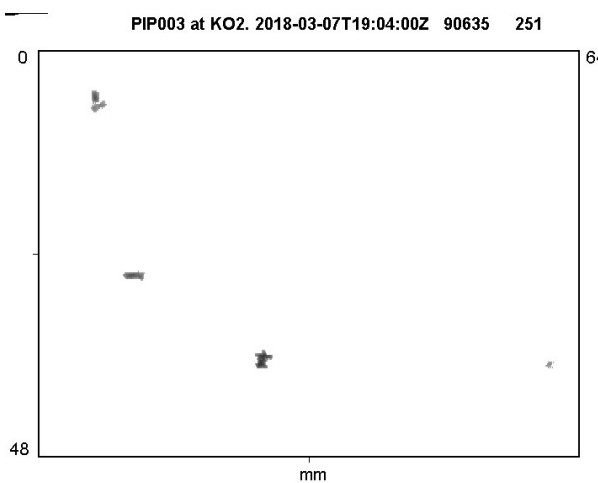

PIP003 at KO2. 2018-03-07T19:04:00Z  90635    251

**Figure B1.** Snapshot taken from a PIP AVI video file collected by the PIP located at the MHS site during ICE-POP. The header can be decoded as follows: 'PIP003 at KO2' indicates which PIP recorded the file, '2018-03-07T12:04:00Z' is the time stamp, '90635' is the 'Q_R' frame index within the current 10-minute period, and '251' is the frame number within this specific AVI file. Note, the black mark in the top-left corner is a digital tag that can be used to link the quicklook image in the AVI file back to the raw data file.

the study and, critically, tested the reprocessed PIP data for quality purposes. Dr. Pettersen provided data support and advice on reprocessing the PIP images.

*Competing interests.*  Some authors are members of the editorial board of Atmospheric Measurement Techniques. The peer-review process was guided by an independent editor. The authors have no other competing interests to declare.

*Acknowledgements.*  The authors would like to thank David Wolff for the fruitful discussions regarding various aspects of this manuscript. Additionally, the authors would like to thank Larry Bliven and Jacopo Grazioli for their insights into the inner workings of the PIP and MASC instruments, respectively. The authors are also indebted to the ICE-POP 2018 field campaign participants for their data collection and processing efforts. Finally the authors would like to acknowledge the efforts of three anonymous peer reviewers whose suggestions have greatly improved this manuscript. Dr. Helms' work was supported by an appointment to the NASA Postdoctoral Program at NASA

Goddard Space Flight Center, administered by Universities Space Research Association under contract with NASA. Dr. Pettersen's work was supported via NASA grants NASA 80NSSC19K0712 and NASA 80NSSC21K0931.

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
