# Peer review of "A Comparative Evaluation of Snowflake Particle Size and Shape Estimation Techniques used by the Precipitation Imaging Package (PIP), Multi-Angle Snowflake Camera (MASC), and Two-Dimensional Video Disdrometer (2DVD)"

_Atmospheric Measurement Techniques, 2021_

## Author Comment (AC1)

======= REVIEWER #1 =======

amt-2021-427: Helms et al.  A Comparative Evaluation of Snowflake Particle Size and Shape Estimation Techniques used by the Precipitation Imaging Package (PIP), Multi-Angle Snowflake Camera (MASC), and Two-Dimensional Video Disdrometer (2DVD)

In short: This study compares the different algorithms behind the measurement techniques of three digital video disdrometers: the Precipitation Imaging Package (PIP), the Multi-Angle Snowflake Camera (MASC), and the Two-Dimensional Video Disdrometer (2DVD) in observing snowflakes. The focus is on defining the uncertainties in the defined area influencing the equivalent diameter, and the aspect ratio. The authors quantify the motion blurring, in the case of PIP also the image compression, the shape-fitting measurements, and in the case of 2DVD, the estimate of the bounding box measurement when particle horizontal motion needs to be adjusted with an unskewing algorithm.

The topic is interesting and relevant for surface observations and the development of retrieval methods in global monitoring of snowfall. The study has novelty in the way it examines the measurement algorithms internal to the instruments, which typically are not transparent to the end-user of the data. The theory is clearly outlined with illustrative examples, and the conclusions are well-supported and valid. The manuscript is well written and provides a clear storyline, however, at least in my opinion, it leaves the reader questioning what is the magnitude/importance of these studied uncertainties in respect to the other uncertainties e.g. particles out of focus or only a fraction of particles observed, wind effects to particle fall velocity, miss-matching of particles, partially illuminated measurement space or limited observations of particles from only one plane projection, these are referred in several publications prior to this one. I would like to see more discussion on this topic and references to other related studies. My recommendation is to publish the paper after addressing this concern and some small remarks mentioned below.

I've added a discussion section that takes a look at how the findings of this study compare to other studies (Section 6, Lines 550 – 587).

Minor comments:

Line 11: "… PIP or 2DVD which provide similar precision once the effects of the PIP image compression algorithm are taken into account." This sentence is somehow unclearly connected to the previous statement in lines 10-11 and is not clear for a reader who just reads the abstract. Please rephase.

I rewrote that bit of the abstract and this sentence was removed.

Line 41: "There are numerous examples of studies which rely heavily on either of these measures of particle size." The statement "numerous examples" follows only by two references, leaving for example relevant fields such as the snow model or satellite retrieval development

unmentioned. I would like to see a broader scan of the research field, just mentioning applications and example references would be enough.

The previous paragraph gives a list of five studies which use one of the two measures of particle size. As such, I've reworded the start of the paragraph that started on Line 41 (and now starts on Line 39) with the intent of making the paragraph serve as a deeper dive into one of the studies. The paragraph now starts off with:

"In the case of Han and Braun (2021), the authors characterize the global three-dimensional distribution of precipitation mean particle sizes using satellite radar data; they used a form of the equivalent diameter as their metric of particle size, specifically the mass-weighted mean liquid-water-equivalent diameter." Lines 39 – 41

Subsequent paragraphs delve into some of the other studies listed (Pettersen et al 2020,2021; Locatelli and Hobbs 1974; Heymsfield and Westbrook 2010).

Line 64: "separate from the snowflake size," This is unclear. I don't understand how these mentioned studies are concerning only aspect ratio separate from size. Please rephrase.

My intent was to make sure the reader wasn't thinking of snowflake shape as referring to the combination of relative dimensions and overall size. In hindsight, I don't think the clarification is strictly necessary, especially if it has the potential to cause confusion. As such, I've simply opted to delete the clause. Line 61

Line 87: Altitude of the site?

Added altitude (789 m) after the latitude and longitude coordinates. Line 94

Line 89-91: I would like to see more data of this event to support the assumptions of aggregation and lump graupels, e.g., time series of temperature, PSD, and mean fall velocity.

I've added a figure showing the PSD, mean fall speed, and the air temperature (Fig. 1). I've also added the following text to the end of the paragraph:

"Although the general presence of these habits were primarily identified through visual inspection of the PIP data, further support of their presence can be found by examining the time series of the particle size distribution, particle fall speed, and air temperature, as depicted in Fig. 1. Periods when aggregates are present can be identified by the larger equivalent diameters and slower fall speeds and periods with lump graupel can be identified by the smaller equivalent diameters and faster fall speeds; the lump graupel can be discerned from liquid precipitation based on the below freezing temperatures, which extend over a deep layer according to a nearby thermodynamic sounding (not shown)." Lines 98 – 103

Line 336-8: "Even with a very fast fall speed of 4 m/s, the overestimation of the equivalent diameter for very large circular particles (diameter ~ 10 mm) is approximately two orders of magnitude or smaller than the actual equivalent diameter." Two orders of magnitude? This is not clear to me.

A fractional bias of 0.01 would mean that the bias is two orders of magnitude smaller than the measurement. In this case, a fraction bias of 0.01 for a circular particle with a diameter of 10 mm would mean the bias is 0.1 mm. I've clarified this point in the text by changing the sentence to read

"For a very fast fall speed of 4 m s$^{-1}$, the overestimation of the equivalent diameter for very large circular particles (diameter $\gtrsim$ 10 mm) is approximately two orders of magnitude smaller than the actual equivalent diameter (i.e., a fractional bias of 0.01)." Lines 389 – 391

Line 339: "perfectly circular". Why assumed the particle to be circular, though written in lines 68-69 "aspect ratio is frequently prescribed, often with a mean value of 0.6 assumed (e.g., Matrosov et al., 2005)" and then without quantification stated that for the oblate particles "the relative (and absolute) effects of motion blurring on the area and equivalent diameter measurements will also grow". Please justify and elaborate.

The choice of using a circular particle comes down to simplicity: a circle is easy to mentally picture, there's no need to worry about assuming an orientation, and the equivalent diameter is equal to the diameter so the "ground truth" is easy to convey in the plots. As for oblate particles, if you make the particle more oblate without changing the area, the number of top-edge pixels will grow, which will increase the area and equivalent diameter biases (i.e., the "absolute effects") as they are more or less proportional to the number of top-edge pixels (as well as the particle speed of course). As we are keeping the area and equivalent diameter constant, the fractional bias (i.e., the "relative effects") will increase as well. I don't feel like this needs quantification to back it up, but it certainly could do with more of an explanation. The text now reads:

"As the particles under consideration become more oblate, with their horizontal dimension growing relative to their vertical dimension, both the absolute and relative effects of motion blurring on the measured area and equivalent diameter will increase relative to circular particles of the same area. An oblate particle of equal area to a circular particle will have more top-edge pixels than the circular particle. Increasing the number of top-edge pixels increases the number of pixels that the motion blur is applied to, thereby increasing the absolute effects of the motion blurring. Furthermore, because the particle area remains constant, the relative effects of the motion blurring are also increasing. For similar reasons, increasing the complexity of the particle outline such that the number of top-edge pixels increases will also result in larger biases due to motion blur." Lines 393 - 400

Figure 4: Could you add the number of analyzed particles and a density plot would add information instead of a scatter plot.

Switched to a 2D histogram and added particle count to color bar title of Fig. 5.

Figure 6. Just to add more information about the particle habit, could the approx. fall velocity be added to the corner of the image? The colored fitted shapes, could the line be slightly thinner or the image larger, it is now hard to see the lines in respect to the shaded image, they are all on top of each other.

I've gone through and added fall speeds for each particle and played around with the line thicknesses of Fig. 7.  Annoyingly, it seems that particles (c) and (d) are not listed in the PIP velocity tables, so I had to go through and manually calculate the fall speeds for these.  I've also made the annotated circles for the MASC and tensor methods thinner to make the PIP annotated circles easier to see.

Lines 404-406. As PIP is only seeing a plane projection of the particle, but here the particle is referred to as an ellipsoid, it is confusing whether in this perimeter stretching factor analysis, the computations are performed in 3D with ellipsoids and is it then assumed the same axes ratio in both directions or is it performed in the 2D projection. Could you please clarify this?

It seems that this is a bit of sloppy word choice on my part.  I've changed ellipsoid to ellipse. Line 464

Paragraphs 408 – 445: I understood that this section provides explanations why the ellipse-fit in PIP has an arbitrary upper threshold close to 0.6, and why with the rectangular fit in PIP, there is a gap in aspect ratio between 0.9 and 1.0. However, it was not always clear, which "gap" the authors were pointing at. I would suggest that you would refer in the text (when addressing for the first time) to the image, where the "gap" is shown. E.g. in lines 436-438, I assume here the authors are referring to Figure 5b?

I've gone through this section of the text and tried to clarify the wording and point the reader to relevant figures more frequently.  Lines 464 – 507

Figure 9. Same as Figure 4. It would be nice to see the number of analyzed particles and then rather a density plot than a scatter plot.

Swapped to 2D histograms and added the number of analyzed particles to the colorbar title for Fig. 10.  I've also removed the height plot based on reviewer 3's comments.

Lines 526-527: "and, as a result, the maximum dimension and aspect ratio measurements are unreliable; however, the PIP variables other than the ellipse and rectangle dimensions appear to be reliable" I assume here it is referred that the PIP-fitted maximum dimension of an ellipse is unreliable and not that the observed maximum dimension is unreliable. Please clarify.

Clarified.  The text now reads:

"Unfortunately, the PIP shape-fitting routines do not perform well on precipitation particles and, as a result, the PIP-fitted ellipse and rectangle dimension (and, therefore, aspect ratio) measurements are unreliable; however, the PIP variables other than the ellipse and rectangle dimensions appear to be reliable." Lines 525 – 528

Lines 528-530: "As the present study has demonstrated, the PIP imagery can be reprocessed and reliable measurements of maximum dimension (the previous comment) and aspect ratio can be made via the application of an alternative ellipse-fitting algorithm, such as the MASC or tensor-based algorithms." In the manuscript, it was described that the AVI file contains only the first 2000 frames from the 10 - minute section, and with 380 frames per second, this translates to 5.3 seconds of data. It is unclear that can an end-user reprocess the whole data volume or just the sample frames in the AVI-files? Could this be elaborated?

The good news is that yes, the entire data volume can be reprocessed from the raw files.  The bad news is that this is a massive amount of data.  My own efforts ended up taking an excessive amount of time to reprocess even a day of the raw data files.  I've been told that the actual PIP processing code is written such that it can be used to reprocess data.  From what I've seen, it should be possible to implement the tensor-based ellipse-fitting method using the IMAQ software so perhaps the data set can be reprocessed that way at some point in the future.

All that aside, I added the following paragraph to the Method section (just after introducing the AVI files for the first time) to make it more clear that the complete data set could (theoretically) be reprocessed:

"Ideally, we would include all PIP images from the 7--8 March 2018 period in our study rather than just those images that are included in the AVI files.  Since the raw PIP image files are archived and contain every frame with precipitation, this is theoretically possible.  That said, due to the very high data rates involved with the PIP data, the processing code would need to be highly optimized or use specialized software packages in order to process even a couple days of the raw data in a reasonable time frame.  Thankfully, the AVI files contain a sufficiently large collection of precipitation particles to enable us to perform our study."  Lines 202 – 207

---

## Author Comment (AC2)

======= REVIEWER #2 =======

This is an important contribution to understanding how various optical imagers observe properties of snowfall. Overall, the manuscript is understandable. While I have no qualms with the included analysis, there does appear to be some low hanging fruit that would greatly improve the impact of the paper. These and other comments are highlighted below.

Specific comments:

Horizontal motion should/needs to be addressed. Sampling/flow issues aside (and this is extremely important but beyond the scope of the article), the open design of PIP will lead to more translational motion vs. the MASC. Based on the field campaign data, there should be surface wind data you could use that would help guide upper-values for analyses like Figure 3. Further, I would like to see how this relates to the PIP vs. 2DVD analysis (e.g. Figure 9).

I've added a 10 m/s particle motion to the analysis and discussion of motion blurring (as well as to Fig 3, now Fig 4). Changes to the text are scattered throughout section 5.1.

The normal caveats apply to studies based on one case during a campaign. Is there the opportunity to conduct this analysis on other cases? How many were available during ICE-POP 2018? More reasoning is needed. While this case provided lots of variety, it is important to demonstrate how varied results are for other types of cases as perhaps you could demonstrate for certain types of cases, the discrepancies between instruments is either amplified or diminished.

I would argue that, for this kind of study, each particle is a separate case, but just to be on the safe side, I processed data from a couple other cases (specifically, 9 January 2018 and 22 January 2018). Looking at the PIP vs MASC vs tensor dimension plots (similar to Fig 4, which is now Fig 5), the differences from the figure in the manuscript are minimal. As such, I've opted to stick with only using the 7-8 March 2018 event to make the narrative simpler. I also added a note mentioning this to the data section:

"Data collected during a snow event on 9 January 2018 and 22 January 2018 were also examined, but are not included here as their inclusion did not produce any notable changes in the results." Lines 103 – 105

The manuscript could be more concise by merging the Data and Instruments sections. For example, the first paragraph in section 3.1 is essentially duplicative with the material under data. I would remove this, then make the sections on the instruments as sections 2.x.

I don't see the duplication between the introductory paragraph of the instruments section (I assume this is what you were referring to) and the contents of the data section. I toyed around with merging the sections, but the result kept feeling awkward and forced so I reverted back to having them as separate sections. In the process of responding to the reviewer comments,

however, I've added the following additional text to the end of the data section that I think helps separate the two instances of "This study…" that might be making the text sound repetitive:

"Although the general presence of these habits were primarily identified through visual inspection of the PIP data, further support of their presence can be found by examining the time series of the particle size distribution, particle fall speed, and air temperature, as depicted in Fig. 1. Periods when aggregates are present can be identified by the larger equivalent diameters and slower fall speeds and periods with lump graupel can be identified by the smaller equivalent diameters and faster fall speeds; the lump graupel can be discerned from liquid precipitation based on the below freezing temperatures, which extend over a deep layer according to a nearby thermodynamic sounding (not shown). Data collected during a snow event on 9 January 2018 and 22 January 2018 were also examined, but are not included here as their inclusion did not produce any notable changes in the results."  Lines 98 – 105

Technical comments:

Line 200: is vs. will be performed?

Switched to 'is'  Line 227

Paragraph (255-268):  This paragraph could be cleaned up. Examples include multiple 'For simplicity' phrase and you could omit 'it should be noted'.  I would lead off with the 2nd sentence to remind the reader, than discuss the number of factors that aren't addressed rather than revisiting the 'number of factors' phrase.

I've gone in and cleaned up the text a bit based on these suggestions.  Lines 283 – 295

Line 275: Agreed, but you should probably provide a citation for this statement. The larger concern is potential horizontal motion

I added a new paragraph to explain the new statistical approach to the motion blurring calculations that I performed for the revision (Lines 296 – 307).  As part of that, I included a sentence explaining our selection of particle motion speeds, including references to Locatelli and Hobbs (1974), Garrett et al. (2012), and Vazquez-Martin et al. (2021):

"The particle motion speeds are chosen as follows (e.g., Locatelli and Hobbs, 1974; Garrett et al., 2012; Vázquez-Martín et al., 2021): 1 m s$^{-1}$ represents a relatively fast falling snow particle, 4 m s$^{-1}$ represents an excessively fast fall speed for snow particles, and 10 m s$^{-1}$ represents relatively fast horizontal motion (albeit below the U.S. National Weather Service wind threshold for blizzard conditions, -15 m s$^{-1}$)."  Lines 300 – 303

Line 304: Extra 'both' in this sentence.
Fixed.  Line 347

Line 321: How about: Motion blur of the top (bottom)- edge pixels occurs when the particle leaves (enters) those pixels during the image exposure period.

This sentence became much simpler after correcting the error Reviewer 3 found where we were double counting the motion blur effects.  The new sentence reads:

"The motion blur of these top-edge pixels occurs when the particle leaves those pixels during the image exposure period."  Line 369 – 370

Figure 3: The sentence starting with 'Calculations… and Specifically…' is repetitive with the body of the text and does not describe the visual properties of the figure. I would omit for brevity or restate in text instead of the caption.

I've removed this part of the Fig. 4 caption (I also remade the figure to be statistically generated rather than the rough estimate I had been making as well as added a 10 m/s line to talk about horizontal blurring).

Figure 4: best fit lines? I would omit the last sentence in the figure caption as this is already included in the text right after Line 355.

Removed final line of the Fig. 5 caption and added "The diagonal grey lines indicate where the x value is equal to the y value."

Figure 7:  Once again, some of the caption is discussed in text (sentences starting with 'This' and the first 'The'.

Removed the sentence starting with 'The' from the Fig 8 caption.  I feel like "This example uses a circle with a radius of 0.5 mm to compute the base perimeter and area" is worth retaining as a brief reminder to the reader.

Line 447:  'Because the 2DVD'

Fixed.  Line 509

---

## Author Comment (AC3)

======= REVIEWER #3 =======

The sudy is an interesting evaluation of processing algorithms to derive two characteristic dimensions, length and width, of snow particles from 2-dimensional images.

Algorithms from three instruments, PIP, MASC, and 2DVD, and their resulting dimensions are evaluated. The conclusions allow future users of these instruments to chose a suitable algorithm.

Only PIP data are used. Using emulated data for testing other algorithms, which are used with MASC or 2DVD data. This provides a fair comparison of the algorithms only. However, this method cannot compare the actual qualities of the PIP, MASC, or 2DVD measurements related to specific instrumental issues. In particular, incase of MASC and 2DVD, the method cannot evaluate if the PIP-derived emulated or the actual measurements (if available) would more accurately represent the particle's dimensions.

The study clearly describes the chosen method and recognizes its limitations.

I suggest publication of the study after some shortcomings have been addressed.

Major issues:

1) All conclusions are vaguely formulated, some are only speculative.

In general the analysis is not sufficiently quantative in comparing the resulting derived measurements from the various algorithms. Similarly, the Abstract uses many "should" and leaves doubts about the usefulness of the conclusions.

Quantify and better describe things like "spread", "agreement", "reasonable estimates", and "greatly underestimates".

Examples:
Improve discussion around Fig.4. In these scatter plots it is difficult to see the differences tated in the discussion of for example 4a) vs. 4b) or 4d) vs. 4e). As suggested by Referee 1, density plots can be more useful. In addition, some quantitative statistical measures will be useful (e.g. to better argue that "MASC-fitted and tensor-fitted ellipses tend to produce fairly similar particle dimensions", L. 394, and that "MASC-fitted ...ellipses tend to provide more reasonable estimates of particle dimensions...", L 396).

I've added mean absolute differences and mean absolute fractional differences to the figure panels (Figs. 5 and 10) and included them in the discussion of the results (Lines 423 – 491, 525 – 549, and some mentions in Section 6).  I've chosen the term 'differences' rather than 'error' as, to me, 'error' implies that I'm comparing against a ground truth, which is not the case.  I've also switched Figs 5 and 10 to use two-dimensional histograms rather than scatterplots as per Reviewer 1's suggestion.

L. 371-372: reformulate this sentence, using a certain aspect ratio will not increase the spread in dimensions.

I've rewritten a large chunk of this section and corrected this bit of poor wording in the process. The replacement sentence reads:

"Using the PIP-fitted rectangle as the basis of the particle dimension measurements produces a sizable improvement to the agreement between the PIP-fitted long dimension and the tensor-fitted long dimension (Fig. 5a,b) but almost no change in agreement for the short dimension (Fig. 5d,e)." Lines 422 – 424

Fig.9: a) not needed, instead argue/discuss that the two heights are the same.; b) quantify the range (and maybe distribution as histogram) of ratios between 2DVD width and PIP width (I guess they vary between 0.8 and 20 and will show two modes on a histogram); c) quantify similarly and then compare to b). That will allow for less vague descriptions than "can sometimes underestimate the bounding box width" (L. 473) or "are surprisingly accurate" (L. 477).

For Fig 10, I've removed panel a and switched to using a 2D histogram rather than a scatterplot (as per Reviewer 1 suggestion) and have added the mean absolute differences and mean absolute fractional differences to the plot panels. I took a look at the ratios and they didn't really add much, but I think the addition of the mean absolute difference has helped reduce the vagueness.

2) The ellipse-fitting algorithm (PIP, MASC, or tensor variant) could also be applied to 2DVD measurements. This can be tested within this study on the emulated 2DVD measurements, adding a valuable aspect to 2DVD measurements. Then, the conclusion about the limited usefulness of the 2DVD measurements can be revisited (L. 519-521.)

I actually did produce tensor-based ellipse fits for the emulated 2DVD images when I reprocessed the data for this study. That said, I made the decision fairly early on to not include an analysis of them in this manuscript (which has had an ongoing problem with ever expanding scope since its inception). As such, I haven't taken a particularly close look at these fits; perhaps a topic for a future paper. That said, I have added a note in the conclusions mentioning the potential application of ellipse fitting to 2DVD images:

 "While not demonstrated here, it may be possible to also implement a shape-fitting algorithm for 2DVD using the reconstructed images captured by the line-scan camera, although the reliability of the resulting shape measurements from such an algorithm would need further investigation to test the impacts of the image skewing." Lines 626 – 629

3) Sect 5.1, L256-258:

"For this study, however, we will examine the theoretical accuracy and precision of the MASC, PIP, and 2DVD area and equivalent diameter measurements in terms of motion blur as determined from the pixel resolution, exposure time (i.e., shutter speed), and particle fall speed."

Motion blur is not related to precision. Overall, I find the discussion around "precison" unnecessary and not well introduced (only later on in Sect 5, L347, it is mentioned what precision or "precise measurement" refers to. The effects of this theoretically higher precision are, however, not discussed in this study. If the authors consider this to be an important aspect of their study, then I would recommend to evaluate the consequences by using the same algorithm with differing pixel resolutions. As MASC measurements are only emulated here, the actual effects of the higher precision remain unclear (and are not part of this study and instrumemnt specific and related to questions such as if the increased precision is accompanied by a corresponding better optical resolution and accuracy).

I should have said "in terms of pixel size and motion blur" as precision is tied to pixel size and accuracy is tied to motion blur. You make a good point in that I don't delve deeply into the precision and I don't make any examination of how precision of the images impacts the measurements of particle dimensions. As such, I've gone through the manuscript and removed most of the mentions of precision. I've kept a few where I felt they were particularly relevant to the discussion.

4) Sect 5.1 and Figures 2 and 3:

The discussion of motion blur and its effects on accuracy seems to be wrong. It is correct that, considering a vertical particle motion during expeosure, the blurring affects both the upper and lower edge. However, the particle extension is not increased (blurred) upwards and downwards. At the start of the exposure, the particle has an upper and a lower edge. Both these edges are moving (blurring) downwards, i.e. blurring will not add extra pixel(s) above, only below. By incorretly assuming added pixels above and below, the authors seem to overestimate blurring by a factor of two.

Thank you for catching this! After thinking about it, you're absolutely correct that we are double counting the motion blur effects. I've gone through and made corrections to the figures and text where needed.

5) The arteficial "cap" is not explained satisfactorily.

Instead, the value of the cap is translated in a certain perimeter stretching factor. However, the authors do not try to explain, why no smaller perimeter stretching factors exist. Assuming that (L. 442-443) only few particles have a smaller perimeter stretching factor seems wrong, I guess (from looking at Fig. 4.g) that there is not a single particle with smaller perimeter stretching factor.

The reason for this cap is likely to be found in effects of pixelation affecting the perimeter by artificially extending it, more noticeably for smaller particles (~0.5mm) than for larger ones.

Having said this, I need to remark that it should be discussed how the perimeter is determined.

L. 410-413: The pixelation effects should be considered.

Fig.7: Reformulating the discussion around the artificial cap may result in that Fig 7 is not needed. E.g., currently the whole discussion about it in L. 418-445 is difficult to understand and doesn't explain the cause of the cap.

Reconsider the usefulness of Fig.8.

I've added a sentence to the discussion of the perimeter stretching factor to note the sources of small increases in perimeter relative to area:

"Small increases in perimeter, such as this, can be introduced by a few very small deviations of the particle edge from a perfect circle as well as by the inability to perfectly represent a circle using square pixels (i.e., pixelation effects)."  Lines 489 – 490

As for figures 7 and 8 (now 8 and 9), I think they materially contribute to the manuscript by enabling a discussion of the sensitivity of aspect ratio to small changes in perimeter length, which is the mechanism responsible for the artificial cap in PIP ellipse aspect ratios.  That said, I have gone through and tried to clean up and clarify the discussion of the perimeter stretching factor.  Lines 464 – 507

Finally, I added a sentence to the instrument description section for PIP that describes the perimeter calculation:

"The IMAQ Vision software package computes the particle perimeter by subsampling the boundary points to produce a smoother representation of the perimeter; for this purpose, the boundary points are located at the corners of the pixels that make up the particle perimeter." Lines 127 – 129

6) Similarily, the apparent gap between aspect ratios around 0.9 and 1.0 is not explained properly (L. 375-376). It seems to stem from the fact that there is a minimum perimeter stretching factor that is above 1 in case a rectangle(square is fit instead of an ellipse/circle. There is no gap, but all particles with smaller perimeter stretching factor are simply "piling up at the aspect ratio of 1.0.

I agree with your explanation, but I would describe such a feature in the distribution as a 'gap'; the "piling up" is simply the cause of this gap.

7) Using ellipses or rectangles that best fit the particle can be used to describe shape, they are, however, not sufficient as complete measurements of the particle's shape. The limitations of the evaluated algorithms could be highlighted better.

Added a paragraph to the introduction that touches on this point:

"As will be discussed in greater detail below, both PIP and MASC produce their measurements by fitting simple two-dimensional shapes (ellipses and rectangles, specifically) to two-dimensional projections of the three-dimensional snowflakes that the instruments are observing. Because snowflakes come in a large variety of shapes, especially when taking aggregate snowflakes into consideration, any attempt to use a simple shape, such as an ellipse or rectangle, to represent these particles suffers from the inherent limitation of under-representing the complexity of the snowflakes. Furthermore, the use of two-dimensional shapes to represent three-dimensional snowflakes adds an additional layer of limitations revolving around the degree to which the two-dimensional projection accurately represents the dimensions and orientations of the three-dimensional particle (e.g., Jiang et al., 2017). Despite these shortcomings, the measurement of snowflake dimensions based on shape fitting has proven to be a useful tool for studying snow microphysics and understanding the relative capabilities of these measurements is critical to their successful use in research and applications." Lines 70 – 79

Other minor or technical issues:

Terminology:

Inconsistent use of terminology:

E.g. "tensor method" only used twice (L179-180 "hereaftyer referred to as the tensor method" and L242 "referred to here as the tensor method"), elsewhere "tensor-fitted ellipse" or "tensor-fitted ellipse method" or "tensor-fitted ellipse measurement"

Or: Inconsistent use of "resolution", not always used correctly. L100 "resolution" refers to the size on the particle that corresponds to one pixel. This is later more adequately referred to "pixel resolution" (e.g. L.269) or "pixel size" (L. 314).

Maximum dimension is not used in this study. The term "maximum dimension" is, however, used three times in the Conclusions. The authors likely wanted to refer to an ellipse- or rectangle-fitted dimension.

I went through and consolidated the 8 or so different ways I refer to the tensor method into a much more manageable number.  Specifically, I now use 'tensor method' to refer to the method in most instances after I first introduce the method in the text.  In cases where I want to be more generic, I refer to it as 'a tensor-based ellipse-fitting algorithm' in order to avoid the question of "what is 'the tensor method'?" before I've had the chance to introduce it (i.e., in the abstract and first paragraph of the conclusions, under the assumption that some readers will start there).

Although, at the end of the paper, I went even more generic and used 'tensor-based algorithms'. When talking about an ellipse constructed using the tensor method or the measurements derived from said ellipse, I use 'tensor-fitted' as the adjective form.

Additionally, I've gone through and switched 'resolution' to 'pixel size' where relevant.

Finally, I've switched 'maximum dimension' to 'long dimension' as I had intentionally avoided 'maximum dimension' up until this point.  The sole exception to this is in the introduction.

Sect 3.3:

Make it clear that the viewing planes are horizontal and that they are separated vertically by 6 mm (or 7?).  Discuss how the "piecing together" of the single line scans is carried out and what errors or accuracies are to be expected. Is the sentence in L. 517-519 ("highly accurate") true? Provide information on pixels and pixel resolution (as done for PIP and MASC in 3.1 and 3.2).

I've updated the first two paragraphs of the section to include the request information.  Lines 156 – 172

L. 199-200 reformulate "made" (measurements are doen or carried out), e.g. "... before the MASC measurements are emulated by using the same ..."

Reworded sentence now reads:

"Because both the PIP and MASC ultimately perform their measurements using two-dimensional images, no special processing is performed to prepare the images before emulating the MASC measurements."  Lines 225 – 256

L.205: "a five pixel particle" is ambiguous as the PIP measured particle image and the emulated MASC image have different pixel resolutions. Use something like "a five PIP-pixel particle".

The sentence now reads:

"In the case of the emulated MASC, a five PIP-pixel particle would have an area of 0.5 mm$^2$ and a maximum dimension of at least 0.3 mm as the PIP pixels are calibrated to be 0.1 mm of each side."  Lines 232 – 233

L. 214: "product of the particle fall speed and the camera observation frequency" seems wrong, should it be v/f?

Not sure why I wrote 'product' here as the code uses v/f…. Either way, the corrected sentence now reads:

"The vertical motion is replicated by shifting the vertical coordinates of the bilinear interpolation upward by an amount equal to the particle fall speed divided by the camera observation frequency; for 2DVD, the camera observation frequency is 68200 Hz."  Lines 240 – 242

Sect. 4.3: Specify that the tensor elements are mean values of the quantities (e.g. square of Delta y) for all particle pixels (or otherwise explain better eq. 1).

I've cleaned up the sentence, which now reads:

"This method works by computing the eigenvalues, $e_1$ and $e_2$, of a two-by-two mass distribution tensor matrix, which is defined as [EQUATION 1], where $\Delta x$ and $\Delta y$ are the distances from the particle centroid in the horizontal and vertical directions, respectively, and the overbars indicate averaging."  Lines 270 – 274

L. 377-378: Repeated use of "expected" and unclear when the increase in aspect ratio (or the period of lump graupel) is.

Reworded sentence and explicitly indicated period of lump graupel:

"It should be noted, however, that the PIP-fitted rectangle aspect ratio does capture the increase in aspect ratio associated with the periods of lump graupel precipitation on 8 March around 0900 UTC and after 1400 UTC."  Lines 436 – 438

L. 389-390: "lack of a warm nose" and its implications should be explained if that is relevant for the discussion.

The presence of a warm nose in the sounding would open the possibility of the circular particles being ice pellets or liquid (or mostly liquid) precip.  The only bearing this has on the discussion is that it supports the classification of particles (c) and (d) being lump graupel.  That said, 'warm nose' might be a bit too niche terminology for this manuscript, so I've changed the wording on the sentence.  The new sentence now reads:

"Based on the relatively circular shapes of the remaining two particles, relatively high fall speeds (black line, Fig. 1), subfreezing near-surface temperatures (red line, Fig. 1), and the lack of an above freezing temperature layer in a nearby thermodynamic sounding (not shown), particles (c) and (d) are likely both examples of lump graupel." Lines 450 – 452

L. 399-401. While it seems intuitively obvious what the sentence tries to explain, it needs to be refromulated for correctness and clarity.

I don't see any issues with correctness with this sentence, but I've tried to improve the clarity. Here's the updated sentence:

"For a particle with complicated outlines, such as particle (b), the particle perimeter is far greater than the perimeter of the ellipse or rectangle of either equal area or equal dimensions." Lines 460 – 462

L. 402-403: remove last part of sentence ("note, extending the short...") to improve clarity.

Removed.

L.473-476: reconsider the explanation, it seems that the example particle should move to the left while moving down to be compressed horizontally.

I believe my explanation is correct since the particle is scanned from the bottom upwards as it falls through the plane of the line scan camera.  As a result, the position of the bottom of the particle remains unshifted while the top of the particle gets shifted in the direction of horizontal motion.  This has tripped me up multiple times, so it certainly warrants clarification in the text. I've added a note to the end such that the new sentence reads:

"This can occur when a particle of sufficiently low aspect ratio is moving in the opposite direction of the tilt of the top of the particle (e.g., a needle crystal whose top is to the left of the particle centroid moving towards the right); this will result in the particle being compressed in the horizontal because the particle is scanned from the bottom upwards as it falls through the plane of the line scan camera."  Lines 536 – 540

L.247 correct spelling: "eingenvalues"

Fixed. Line 274

L. 304 delete duplicate "both"

Fixed. Line 347

---

## Referee Report (RR1)

**Review comments on amt-2021-427-author_response-version3**
2022-10-22

Thank you for considering my comments and your changes to the manuscript. I agree that removing the analysis of the size aspect from the manuscript seems unfortunately the best solution. You are right that a fair comparison of sizing is difficult on emulations and would require a true instrument comparison, which isn't easy to do.
With these changes, I think the conclusions from the remaining comparison of shape estimation techniques stand out more clearly.
I only have a few suggestions for small modifications (see below points *1-5*) prior to publication. In addition, I am reporting my reflections on your response about the aspect ratio capping.

*1)*
In Sect. 3.1:
Delete the sentence "Furthermore, IMAQ uses one corner … pixel center."  You already specify that corners are used two sentences earlier.

*2)*
In Sect. 4:
Your changes in L184-202 explaining that PIP image processing is now applied to the raw PIP images PRIOR to applying the shape-fitting algorithms are very useful. You can improve them by making clear that this happens before emulation and fitting. E.g., insert a "prior" or "first" in an appropriate place.

Reflections on your response to my previous comment 3. Artificial cap:

Thank you for testing the perimeter stretching factor of circles.
Three questions to make sure I understood your procedure:
Circle: "based on the distance of each pixel center from a randomly perturbed center … spatial ant-aliasing". What is the result of this, B&W "pixelized" image of circle?
Perimeter: IDL contour function to produce contour (=perimeter) at value of 0.5 or 1. I assume that with these values you refer to the pixelized image with value 1 for any circle pixel and 0 for any outside pixel.
How have you determined area?

The value of 0.5 would be the obvious choice (similar to choosing 0.5 as threshold for images blurred between 0 and 1). With that value the perimeter stretching factor has a value close to the minimum value that I expected based on the observed "cap". [The contour for the value 1 is of course smaller. The size dependence is not surprising. I cannot comment more as I don't know how you have determined area.]
Thus, your perimeter-stretching-factor test with circles seems to confirm that "pixelation" alone would be sufficient to explain the pronounced "cap". However, as you have applied a different method to your "pixelized" circles, we cannot draw any firm conclusion. With different method

I mean that you have used the IDL contour function to determine the perimeter (and something to determine area) instead of the method used in the paper, i.e. using the IMAQ area and perimeter.

*3)*
As we cannot conclude anything concrete from your testing, I think the new section in L 350-355 (In theory, … algorithm) doesn't add value to the paper. I would just delete it or replace it with something like (not as new paragraph):
"As the details of the IMAQ perimeter calculation are not well documented, it remains difficult to determine the exact influence of pixelation effects on the PIP shape fitting algorithm in case of perfect circles."
*4)*
In the preceding sentence (L347-349) you may consider including a parenthesis:
"Small increases in perimeter, such as this, can be introduced by a few very small deviations of the particle edge from a perfect circle (DUE TO AN ACTUAL NON-SPHERICITY OF THE PARTICLE OR DEFECTS IN THE IMAGE QUALITY) as well as by the inability to perfectly represent a circle using square pixels (i.e., pixelation effects)."

As you have acknowledged, there is more to it, e.g.:
Pixelation may affect the other instruments too (only that its effects are more pronounced for PIP ellipse fitting).
The PIP image processing may also affect the perimeter.

*5)*
In Conclusions:
Consider splitting the second bullet point (L447-451) in two bullet points. The first one would be about "overestimating the long dimension and underestimating the short dimension". The second one would be about the high sensitivity to small perimeter changes. Both points are important conclusions. Both taken together make the results from PIP ellipse fitting unreliable, as you have discussed.

---

## Editor Decision (ED1)

After reading the revised version and exchanging with the reviewers, I have some concerns that should be addressed before the manuscript can be accepted for publication in AMT.

I have exchanged with Reviewer 3, and I would like to share with you some of his points. To summarize, the question of the pixel size (related to the PIP compression approach) still needs some clarifications, as well as the influence of motion blurring on the derived particle size. These are items 1 and 2 below, that should be addressed. A 3rd item is listed and should be discussed, even if briefly only.

I copy below the detailed comments from Reviewer3 (concerning the last revised version).

**1. Pixel size and compression**

Re "average of 1.01 pixels added", which now is "1.0 pixels when suspended based on our statistical analysis" (L359 in ATC2-1)

I assume this could or should read "1.0 pixel on average when ...".

However, it is still not clear where the 1.0 pixel added on average comes from.

If this value is 0 or 1.0, depends on how the pixels are averaged during compression. As the authors claim that 1.0 pixels are added based on their statistical analysis, I assume that the PIP compression use the second way (of the two that I describe below) of averaging during compression. This is likely a result of the algorithm used. It should be clearly explained that is, apparently, due to this algorithm that the particle appears larger on compressed PIP images.

Two ways of averaging during compression; one using actual averaging of the cover percentages, another one using an assumed two-step algorithm:

One way of averaging would be to take the mean of the two cover percentages as the cover percentage of the new, compressed (0.1mm x 0.2mm) pixel. In that case, a 4-pixel (uncompressed or decompressed pixels of 0.1mm x 0.1mm) particle would after compression and averaging be still 4 pixels (0.1x0.1). Only in one special case 2 pixels (0.1x0.1) would be added by averaging, and this special case is the one depicted in Fig. 3a), third column, where the particle covers two compressed pixels (0.1x0.2) fully and two partially at 50% with one uncompressed pixel (0.1x0.1) covered (100%) and one uncovered (0%). If starting from this special case, the particle is moved somewhat up or down, then of these two partially covered compressed pixels (0.1x0.2) one will assume a cover percentage of larger than 50% and one smaller than 50%, i.e. only one of the two will be included in the particle image resulting in a 4-pixel (0.1x0.1) particle image. In other words, apart from the special case, all cases result in no additional pixel and 0.0 pixel are added, not 1.0.

Another way of averaging would be to first convert the uncompressed pixels (0.1x0.1) to 100% or 0% depending on the cover percentage being >= 50% or <50%, respectively. Then, in a second step, the mean of two of these 100% or 0% values is taken as the new averaged compressed (0.1x0.2) pixel. As an example (see FIGURE), if the particle covers 70% of one and 0% of the other uncompressed (0.1x0.1) pixel of the two that make up one compressed (0.1x0.2) pixel, then in the second step one becomes 100% and the other 0% and with that the compressed (0.1x0.2) pixel becomes 50% and is counted.

[Figure]

FIGURE: A 4-pixel one-dimensional particle placed on a grid of uncompressed (0.1 mm x 0.1 mm) pixels so that the top-most part of the particle covers 70% of an uncompressed pixel and the bottom end covers 30% of an uncompressed pixel. One way of averaging pairs of uncompressed pixels to get cover percentages of compressed (0.1 mm x 0.2 mm) pixels is indicated on the left, another way on the right.

**2. Motion blurring**

In the analysis in Sect 5.1, the authors argue that any point on the particle is blurred into a line along the direction of motion. The length of the line is calculated from the speed and exposure time. This length is added to the particle extension and results in a corresponding addition of pixels in the image according to the statistical analysis of randomly placing the enlarged particles on the pixel grid.

On a particle picture with sufficient quality, the particle would indeed appear extended by one time the motion-blurring length. It would appear "blurred" at both the top and bottom edges (in case of vertical motion). The blurring at these two edges would occur over one motion-blurring length at each edge. If and by how much the particle size will be modified by this motion blurring, depends on how the particular instrument "takes" the image of the blurred particle. I am referring to the technique or algorithm that decided which pixel belongs to the particle or not. In case of a suitable threshold discriminating background from particle pixels, the particle size will result unchanged. Depending on the exact threshold value, the particle could be smaller by up to one motion-blurring length, larger by one motion-blurring length. Or anything in between. It appears that the authors have implicitly chosen a very sensitive threshold that accounts to the particle everything that has the slightest difference to background and, in this way, enlarges the particle by one motion-blurring length.

Thus, this method of determining the additional pixels by adding one motion-blurring length to the particle without considering instrument specific thresholds or techniques may clearly over-estimate the measured size. The inevitably diffuse edges of the optical image (due to motion blurring and also other effects of optics or particle location within the sampling volume) will result in a sizing accuracy or bias. The particle enlargement presented in Sect 5.1 represents the maximum contribution that motion blurring can have on this sizing accuracy or bias. Depending on the instrument, the actual effect of motion blurring may be smaller. This should be discussed and the amount the effect will likely have be evaluated for each instrument, MASC, PIP, and DVD.

There is a last point that could be clarified, related to

**3. Artificial cap and discussions around Fig 7 and Fig 8**

I still believe that the main reason for the artificial cap is primarily due to pixelation effects and not due to the sensitivity indicated in Fig 9.  That the cap is so pronounced is, as pointed out in the discussion of Fig 9, a result of the sensitivity to changes in the perimeter stretching factor when it is close to 1.

I would suggest that the authors do a new statistical analysis of circles being pixelated. Circles of various sizes are placed randomly on pixel grids and are then pixelated using a suitable algorithm (e.g. each pixel covered by more than 50% by the circle is accounted as particle pixel). From the analysis of these pixelated images a histogram as in Fig 9 can be plotted. Clearly, for a circle one would expect the perimeter stretching factor to be 1. Due to pixelation, however, I expect the perimeter determined from the pixelated image to be longer than the actual circle perimeter. Consequently, the perimeter stretching factor will be larger than 1 and, in the histogram, it will likely have a peak at a certain value (just as in Fig 9 now). There will also be a minimum perimeter stretching factor (which will result in an artificial cap if the same data are plotted as in Fig 5g). I would not be surprised if this minimum stretching factor would be around 1.05, i.e. explaining the artificial cap at 0.65.

---

## Author Response (AR2)

Responses to Reviewer 3 Revised Manuscript Comments

The manuscript has improved significantly as a result of the authors responses to the referee comments. In particular the analysis has improved and details like density plots (2D histograms) in Figs 5, 10, as well as using mean difference as measure for comparison.

Some of the modifications include major changes and editing. In the following I have several remarks, comments, and questions regarding some of the changes. Please have a look and change and improve/clarify the manuscript.

Unless otherwise noted, i refer to Fig. and Lines as they are numbered in the version2 manuscript.

RE Major 3) (Precision)

L404-405: "As we established above, the 2DVD is the least affected by motion blur and, as a result, has the most accurate area measurements, although the large pixel size may impact the precision of those measurements."

I still beleive that "precision" is the wrong term to use here.

Larger pixels ultimately affect accuracy of determined dimensions (picel size will contribute to the error and this contribution will scale with the pixel size). Furthermore, the negative effects of pixelation will be intensified with larger pixels (with consequences on perimeter stretching, see below about "cap").

While I'm sure we could come to a consensus on this eventually, I've opted to remove it since it's not particularly critical to the paper. The new sentence reads:

"As we established above, the 2DVD is the least affected by motion blur and, as a result, has the most accurate area measurements, although it should be noted that 2DVD also has a relatively large pixel size." Lines 409 – 411

RE Major 4) Motion blur

Corrected. Added horizontal blur at 10m/s? Check new Fig 4 and discussion.

MASC 33.5um one pixel; 40us exposure

=> at 1m/s 40um blurred "streak" for any arbitrary point on particle This blur streak is linearly (if motion along one pixel axis) on average on 40um/(33.5um/pixel)= 1.19 pixels.

This expected value is equal to that found by statistical analysis of 100000 random particles.

New L311-312: "would contribute to at least two pixels and possibly three, although our

statistical analysis indicates an average 1.19 extra particle pixels are added." Regarding contributions to three pixels, it may be worth commenting: ...three: in that case the contributions to two of these pixels is minor.

Mention that 1.19 is exactly as expected!

Corresponds to the average extra pixels you show in brackets for 4m/s and 10m/s

Added note that the 1.19 is as expected and note about the contributions being minor in the three pixel case:

"With a pixel size of 33.5  $\mu$ m, there would be minor motion blurring as a single point on the particle would contribute to at least two pixels and possibly three, although the contribution to two of the pixels in the three-pixel case would be minor. Our statistical analysis indicates an average 1.19 extra particle pixels are added, which matches the expected value that can be calculated from the motion and pixel size." Lines 314 – 317

L315: "For a particle moving at 10 m s-1..."

You seem to mean "For a particle moving horizontally at 10 m s-1..." Note, the 400um blurred line is true for a vertically or horizontally moving particle (for normal camera orientation), not for any other oblique angle (i.e. the most likely motion). Nevertheless, I would not change the analysis by including oblique angles (would make it unnecessarily complicated).

I added "horizontally" as suggested (technically it also applies vertically, but that shouldn't be happening under normal circumstances) as well as a note that the analysis doesn't apply to oblique angles:

"For computational simplicity, these calculations assume that a pixel is part of the particle if it has at least 50% coverage and that the motion is not oblique to the pixels (i.e., only completely vertical or horizontal motion is considered)." Lines 309 – 311

"For a particle moving horizontally at 10 m s-1, the point appears as a 400  $\mu$ m line, contributing to at least 12 pixels (average of 11.94 extra pixels), although it should be noted that the enclosed sampling volume used by MASC would make such a particle motion less likely than it would be with an open sampling volume." Lines 320 – 322

Is motion blur in horizontal and vertical directions treated separately for PIP (L319-321)?

Yes. The horizontal motion blur will be the pre-compression blur while the vertical blur will be the post-compression blur. To clarify this, I added a note:

"For horizontal motion, corresponding to the pre-compression blurring effects, PIP has a pixel size of 0.1 mm (100  $\mu$ m)." 326 – 327

L353-356: "Since the image compression only impacts vertical motion blurring, the 10 m s-1 particle motion is not included in Fig. 3, but were a snowflake to fall at such an extreme speed, motion blurring would, on average, add 3.80 pixels

after accounting for the effects of image compression." "the 10 m s-1 particle motion" => "the horizontal 10 m s-1 particle motion"

I feel like this might cause confusion as the rest of the sentence is dealing with vertical motion.

"add 3.80 pixels after accounting for ..." => "add 3.80 pixels after accounting for ... and X pixels without compression (for horizontal motion)"

To match the wording in the rest of the paragraph, I change the wording of the sentence to read:

"Since the image compression only impacts vertical motion blurring, the 10 m s-1 particle motion is not included in Fig. 3, but were a snowflake to fall at such an extreme speed, motion blurring would, on average, add 2.80 pixels before accounting for the effects of image compression and 3.80 pixels after accounting for image compression." Lines 360 – 362

Fig 3 caption: "The darker blue shading indicates the area covered by the actual particle at the end of the image exposure period" => "... the start of the image exposure period"?

Since the particle is falling, the actual particle (darker blue shading) is located at the bottom, so I think "at the end of the image exposure period" is correct.

L 351, PIP: for suspended particles an average of 1.01 pixels added??

The 1.01 have not been mentioned or explained. (Maybe thsi refers to L345-346 ("the image compression can add between zero and two additional pixels to the height of a suspended particle.")

Comment: Even without pixel compression: a four pixel particle may result in a five pixel particle on picture (50% on lowest and 50% on highest pixel.

It appears that there was a minor statistical fluke; there should only be an average of one additional pixel. The compressed pixel values are the primary reason I went with a statistical approach rather than just computing an expected value as the compression effects are not straight forward to account for. The rest of the additional postcompression PIP pixel counts appear to be accurate. The sentence now reads:

"Because of our prescribed threshold of 50% coverage, the 1-m-s-1 fall speed case ends up producing the very similar results to the suspended particle case: the addition of between zero and two additional pixels to the height with an average of 1.28 extra pixels at 1 m s-1 compared to 1.0 pixels when suspended based on our statistical analysis" Lines 355 – 357

L366-372: The area error also depends on particle orientaion. A needle or column is more affected if oriented horizontally.

This is correct (see lines 372 – 373 of newly revised manuscript). I also considered adding a parenthetical note after shape to suggest habit as an example, but I decided against it as aggregate shape is also very important and I don't want to mislead the reader into focusing on habit alone.

RE Major 5) "Cap" Major 6) gap

I think you should discuss in appropriate places the problematic definitions of the PIP ellipse and rectangle and the consequences of these: long dimension of ellipse can be far longer than maximum dimension; pixelation leads to lowered aspect ratio for elipse; many particles for which rectangle fit is not possible. These problems explain the "cap" in elipse fitted aspect ratios and the apparent gap (in rectangle-fitted aspect ratios).

While I do discuss these topics throughout the paper (Appendix A goes in depth into the precise definition of the PIP ellipse and rectangle; Section 5.2 covers the problems with the shape fitting that result from how the PIP shapes are defined), I do agree that they should also appear in a brief summary at some point to remind the reader. To this end, the relevant bullet point in the conclusions section now reads:

"The PIP shape-fitting algorithms do not perform well due to their reliance on only the area and perimeter of a particle leading to a tendency towards overestimating the long dimension and underestimating the short dimension that is highly sensitive to small deviations from the smooth-edged fitted shape; this renders the following variables unreliable: `Ellip\_Maj', `Ellip\_Min', `Rec\_BS', and `Rec\_SS' (section 5.2); other variables reported by the PIP are unaffected by the shape-fitting issues." Lines 615 – 619

L621 "PIP suffers from the issue that the shape-fitting routines do not perform well on precipitation particles..."Where was this discussed in paper? Does it refer to the "cap"/pixelation discussion (see below)? Related to my comments

on problematic definitions of PIP ellipse and rectangle (see above)? In any case, it is unclear to me why you talk about precipitaion particles here. Do you mean ice/snow?

The performance of the PIP shape fitted routine is covered in Section 5.2. Regarding "precipitation" I would argue that the PIP shape-fitting will not do well with fitting an ellipse to the slightly distorted circle of a rain drop either (see Fig. 8a), but since this paper is explicitly about ice/snow, I've added the word "frozen" before "precipitation" to clarify this. Line 631

L623: "PIP variables"?

I've reworded a couple sentence to clarify this:

"Worth noting, however, is that the PIP shape-fitting issues only impact the four variables listed in the relevant bullet point above; the other PIP reported measures of particle size (e.g., particle bounding box dimensions, area, equivalent diameter) remain

reliable. As such, any products that use a non-shape-fitting measure of particle size (e.g., Pettersen et al., 2020) remain unaffected by the shape-fitting issue." Lines 632 – 636

Please explain what you mean with "cap" when you first use this expression!

New L432-434: "In terms of aspect ratios, the PIP-fitted rectangle aspect ratio (Fig. 5h) does not suffer from the artificial cap that was present with the PIP-fitted ellipse aspect ratio (Fig. 5g) and much of the reduction in the mean absolute difference (0.254 versus 0.184) is likely due to this change."

I've added the following sentence to the end of the paragraph where we point out the artificial cap in the PIP-fitted ellipse aspect ratio (although we didn't refer to it as an artificial cap there):

"This effectively represents an artificial cap in PIP-fitted ellipse aspect ratio." Line 427

L432: Cap? Fig 5h shows rectangle which does not have the "cap", refer to Fig 5g) instead!

I think the figure references are correct as they stand. I refer to 5h as the PIP-fitted rectangle aspect ratio, which I point out does not suffer from the cap, and 5g as the PIP-fitted ellipse aspect ratio, which I point out does suffer from the cap.

L 478: refer to Fig 6a): I would refer to Fig 5g) and Fig 6a).

Added reference to Fig 5g:

"The lack of shape information in the PIP shape-fitting equations does not entirely explain the artificial cap in aspect ratio at ~0.6 for the PIP-fitted ellipses (as in Figs. 5g and 6a), however." Lines 485 – 486

L434: What is "this change"?

Poor wording on my part. Here's the new sentence:

"In terms of aspect ratios, the PIP-fitted rectangle aspect ratio (Fig. 5h) does not suffer from the artificial cap that was present with the PIP-fitted ellipse aspect ratio (Fig. 5g) and much of the reduction in the mean absolute difference (0.254 versus 0.184) is likely tied to the lack of said artificial cap." Lines 438 – 440

L434-435 "That said, the PIP-fitted rectangle aspect ratio still has a tendency to greatly underestimate the aspect ratio relative to the tensor-fitted ellipse aspect ratio (Fig. 6b)" Cannot be seen from Fig 6b (alone) => Refer to Fig 5h!

Added reference to Fig 5h and Fig 6d:

"That said, the PIP-fitted rectangle aspect ratio still has a tendency to greatly underestimate the aspect ratio relative to the tensor-fitted ellipse aspect ratio (Figs. 5h

**and 5b,d)." Lines 440 – 441**

L436-438: "It should be noted, however, that the PIP-fitted rectangle aspect ratio does capture the increase in aspect ratio associated with the periods of lump graupel precipitation on 8 March around 0900 UTC and after 1400 UTC."

The text suggests that you are looking at Fig 6b. It is not obvious at first waht you are referring to exactly: the higher density at higher aspect ratios. But then you should say the same also for aspect ratios from PIP-fitted ellipses and refer to both Fig.s 6a and 6b).

The intent here was to point out that the PIP-fitted rectangles capture the large aspect ratio while the PIP-fitted ellipses do not; now that you point it out, I see the choice of wording I used was suboptimal since both show an increase in the aspect ratio. As such, I've reworded the sentence to clarify this and to make it clear that the periods of lump graupel correspond to periods where the 1.0 aspect ratio is dominating. I also swapped the order of this sentence and the previous one to improve the flow:

"In contrast to the PIP-fitted ellipse aspect ratios, the PIP-fitted rectangle aspect ratio does capture the large aspect ratios associated with the periods of lump graupel precipitation on 8 March around 0900 UTC and after 1400 UTC, although these aspect ratios are almost entirely reported as a value of one. Interestingly, the PIP-fitted rectangular aspect ratio frequently has a value of one but very rarely has a value between 0.9 and one; this peculiarity will be revisited later in this section." Lines 441 – 445

L447-454 Discussion of Fig 7: example particles.

You should state that you now refer to "MASC-fitted and the tensor-fitted ellipses" as correct or reference.

I'm hesitant to declare one (or both) of these as the "correct" measurement, hence the "more reasonable estimate" wording in the final sentence of this paragraph:

"Additionally, visual examination of these and other (not shown) individual particles suggests that the emulated MASC-fitted and the tensor-fitted ellipses tend to provide more reasonable estimates of particle dimension than either of the PIP-fitted shapes." Lines 463 – 465

As for being used as a reference, I feel like the last sentence of the first paragraph of section 5.2 covers that sufficiently:

"Although the tensor-fitted ellipse measurements will be used here as a point of comparison between the various instrument algorithms, it should be noted that the tensor-fitted ellipse measurements are not a `ground truth' and are subject to errors of their own." Lines 417 - 419

Panel c and d: L449-452 "Particles (a) and (b) are both likely some type of aggregate frozen precipitation based on their odd shapes.

450 Based on the relatively circular shapes of the remaining two particles, relatively high fall speeds (black line, Fig. 1), subfreezing near-surface temperatures (red line, Fig. 1), and the lack of an above freezing temperature layer in a nearby thermodynamic sounding (not shown), particles (c) and (d) are likely both examples of lump graupel."

The particles are named after the panel letter of the unannotated view of the particle, hence "particle (a)" is the particle that appears in panels a and e, "particle (b)" is the particle that appears in panels b and f, etc.:

"For simplicity, these particles will be referred to by the Fig. 7 panel letter of the unannotated panels (i.e. panels a--d)." Lines 455 – 456

L460-463: "For a particle with complicated outlines, such as particle (b), the particle perimeter is far greater than the perimeter of an ellipse or rectangle of either equal area or equal dimensions."

This is wrong: the definition of the fitted ellipse is that it has the same perimeter and area, so you cannot say that the perimeter of the particle is far greater than that of the ellipse. Further, it is unclear what "equal dimensions" refer to.

Good catch! Not sure what "equal area" is doing in there (presumably there was a reason, but it's escaping me now; best guess is I was editing this sentence after editing the following paragraph where we keep the area constant). It should just be "of equal dimensions." On the topic of dimensions, I've adjusted the wording to clarify things:

"For particles with complicated outlines, such as particle (b), the particle perimeter is far greater than the perimeter of an ellipse or rectangle of equal dimensions (i.e., length and width)." Lines 467 - 469

L464: "perfect ellipse"??

"The relationship between excess perimeter for a given area, relative to a perfect ellipse..."

It is the excess with respect to a circle (not "perfect ellipse".

The choice of "perfect ellipse" here was intended to clarify that we are not talking about an ellipse that has been pixelated nor something that is very closely approximated by an ellipse but deviates slightly from a "perfect ellipse". Ellipse is preferable to circle as it's the more general case and the concept that the sentence is introducing can be readily applied to a non-circular ellipse as well as to the special case of the circle. We use a circle here instead of an ellipse of some arbitrary aspect ratio as there is no analytical solution for the perimeter of an ellipse in terms of major and minor axis length.

L475-476: "this missing information is, in fact, the core issue with both the PIP ellipse and rectangle fits."

This is one issue, i.e. the definition of ellipse and rectangular fits. The other issue is pixelation.

circle should result in perimeter stretching of 1 and aspect ratio of 1, but it does not due to pixelation.

The pixilation issue, while important, is secondary to the lack of shape information in the fitting equations. If the relationship between perimeter stretching factor and aspect ratio (Fig 8a) were linear instead of exponential (i.e., the fitting equations were better), then pixilation would not have as large of an impact on the ellipse fit. I would argue that having the shape information in the fitting process acts to reduce the sensitivity of the fit to pixilation errors. Evidence of this can been seen in the various comparisons between the PIP-fitted shapes (especially the ellipse fit) and the MASC and tensor fitted ellipses (both of which do take shape information into account). Since all the shape fits use the same images, they all encounter the same level of pixilation, but the fits that include shape information produce far more reasonable estimates of the particle shapes. Looking at the opposite situation, even with zero pixilation, slight deviations from a perfect ellipse will still result in relatively large changes to the ellipse for all but the smallest aspect ratio particles (as per Fig 8a).

L489-490: "...very small deviations of the particle 490 edge from a perfect circle as well as by the inability to perfectly represent a circle using square pixels (i.e., pixelation effects)."

Cannot separate these effects. However, pixelation alone explains the "cap" regardless if there are any small natural dieviations (that are not resolved due to the same pixelation.

I agree that you cannot separate the small deviations from the pixilation effects in practice, but they are not synonymous. The pixilation effects create small deviations, but small deviations are also introduce by the actual true shape of the particle being observed. Whether or not these small deviations are captured in the pixelated image would depend on how they line up with the pixels, the size of the deviation relative to the pixel size, and how the deviation interacts with the image compression (which introduces a directional dependency to the pixilation effects).

Take a circle and determine the measurements after emulating PIP images. Then check the perimetr stretching factor. You will likely get a perimeter stretching factor larger than 1 as a consequence of pixelation and determination of the perimeter of the pixelized particle image. The pixelation itself creates a variation of perimeter location from any defined centre point around the radius of the equivalent circle. Cfr. Fig. 9.

Outside of some very unique circumstances involving perfect alignment of the pixels and the circle, I agree with this statement.

Rectangle fit: perimeter stretching below 1.128: apparently the algorithm does a compromise and accepts a square with too long perimeter and too small area (and aspect ratio of 1).

The rectangle is less susceptible to pixelation: the "cap" therefore does not exist.

The lowest perimeter stretching factor (~1.05, see below) is below the limit where rectangular fits are possible. Therefore, no cap.

Discussion in L499-507 can be simplified, it is all due to the definition of rectangle fit.

Figure 8 indicates that the ellipse and rectangle fit are similarly sensitive to small changes in perimeter stretching factor (i.e., small deviations from a perfect circle, or, more generally, a perfect ellipse), but, critically, this sensitivity occurs at different perimeter stretching factors. The lack of a cap is primarily due to where the distribution of observed particle perimeter stretching factors peaks relative to that sensitivity (see Fig. 9 and related discussion). If the observed perimeter stretching factor distribution were to be maximized at one, then we wouldn't see the artificial cap in the PIP-fit ellipse (it wouldn't make the fits any better, it would just mean that the distribution would look deceptively reasonable and potentially mislead researchers to trust the data).

Fig 9 shows again that rectangle fit is not well defined: for half the particles the fit is not possible and the PIP algorithm assigns an aspect ratio of 1.

This is precisely why there does not appear to be an artificial cap in aspect ratio and why we argue that, while the distribution of PIP-fitted rectangle aspect ratio might look reasonable at first glance, the truth of the matter is the PIP-fitted rectangles do not produce reliable measures of shape either.

There is nothing special about the aspect ratio 0.6. Instead you could look at tha maximum aspect ratio for PIP-fitted ellipse that you found (~0.65) and relate that to the minimum stretching factor found for PIP (~1.05).

You are correct that the aspect ratio of 0.6 is more or less arbitrary (at least among values that are close to the maximum PIP-fitted ellipse aspect ratio). There's no real narrative difference between using 0.6, using the maximum PIP-fitted ellipse aspect ratio (which is approximately 0.65, as you pointed out), or using a value of 0.525, which approximates the peak of the distribution during the lump graupel periods.

Fig 7 improved in New Fig8; check discussion! Check discussion new Fig 8 and 9 (tracked changes new L 464-507: perimeter stretching)!

I appreciate the new sentence about PIP perimeter calculation (Lines 127 – 129), but it is not clear:

1) boundary points are located at the corners of the pixels that make up the particle perimeter

2) subsampling the boundary points to produce a smoother representation of the perimeter

What does subsampling mean?? How does smoothening work?

This is crucial: the smoothening and perimeter calculation affects the perimeter stretching factor and enlarges it more or less (the pixelation effect).

Here's the entirety of information I have on the perimeter measurement that the IMAQ software provides (from the 2003 version of the IMAQ manual):

"Perimeter: Length of a boundary of a region. Because the boundary of a binary image is comprised of discrete pixels, IMAQ Vision subsamples the boundary points to approximate a smoother, more accurate perimeter. Boundary points are the corners of the pixels that make up the boundary."

Personally, I interpret this to mean that the perimeter of a corner pixel will be taken along the diagonal of the pixel rather than around the edge of the pixel. Either way, I've clarified this by tweaking the sentence and adding another (as well as pointing to the 2003 manual since the 2000 manual doesn't have this information):

"According to the 2003 IMAQ Vision software manual (National Instruments, 2003), the IMAQ Vision software package computes the particle perimeter by subsampling the boundary points to produce a smoother representation of the perimeter; for this purpose, the boundary points are located at the corners of the pixels that make up the particle perimeter. Our interpretation of this is that a corner pixel (i.e., a particle pixel sharing exactly two adjacent sides with other particle pixels) will contribute only its diagonal length to the perimeter rather than the length of the exterior two sides. How this subsampling behaves for a particle pixel that only shares a single side with other particle pixels is unclear." Lines 127 - 131

**MINOR:**

L 299: "...particle moving at ... 10 m/s (in the along-particle direction)" Is the "along-particle direction" the direction defined by the particle motion? Then that is not needed. In any way "along-particle direction" is confusing. The along-particle direction is the direction along the long-axis of the particle. Reworded sentence to avoid confusion:

"To simulate the particle motion blurring, we generate 196 one-dimensional particles, whose lengths are uniformly distributed between 0.5 mm and 20 mm, and then add the expected motion blurring for a particle moving at 1 m s-1, 4 m s-1, and 10 m s-1 in the direction of the particle's long axis for each instrument." Lines 300 - 302

Sect 5.1: movement speed motion speed fall speed I think "movement speed" and "motion speed" sound somewhat odd. I would use "speed" and/or "motiion". Further, "horizontal movement of particles" => "horizontal movement of particles" sounds better. I think movement speed is more explicit than motion, although velocity is an even better term. I've gone through and made the terminology more consistent by using velocity. The goal behind "particle movement speed" was to reduce the chances that the reader misinterprets "particle speed" to be only the fall speed (which is only one component of the particle velocity, the other being the horizontal speed)

Sect 5.2 measurements made using => measurements emulated

I was originally going to swap these out, but doing so makes the sentence inaccurate as neither the PIP measurements nor the measurements based on the tensor-fit ellipse are emulated (i.e., the tensor-fit ellipse is not emulating another instrument). Upon further consideration, we are making measurements of the particles, those measurements are emulating the measurements of the MASC and 2DVD. Thus, both "measurements made using" and "measurements emulated" are technically correct for the MASC and 2DVD methods although "measurements emulated" give a more specific description for the MASC and 2DVD methods here.

L467 duplicate "area... and area"

Fixed

L601: "...although the relatively low resolution may impact the precision of those measurements (section 5.1)."

Also here, I would make clear what you mean with "resolution":

"...although the relatively low resolution (large pixel size) may impact the precision of those measurements (section 5.1)."

See also the comments on precision above.

See my response to your first comment. Changed wording to read:

"The 2DVD camera setup provides the most accurate measure of area (and equi-area diameter) due to its resilience to motion blurring, although it should be noted that 2DVD also has a relatively large pixel size (section 5.1)." Lines 608 – 609

---

## Author Response (AR3)

Response to editor comments:

I've spend a couple weeks implementing and then troubleshooting some code to test the effects of motion blurring on the particle area measurements using circular particles while taking the individual image processing algorithms of each instrument into account. While I was successful in implementing this I found that applying a 50% threshold (2DVD uses a flat threshold determined via calibration) to the motion-blurred circular particles resulted in the complete removal of the motion blurring effects. At first I found this rather confusing, but, as I thought about it more, I came to realize that this was not a coding issue but a fundamental flaw with this approach to examining the motion blurring. Motion blurring of a step function effectively replaces the step with a linear gradient. The motion blurring algorithm I had implemented (specifically, Korein and Badler 1983) assumed the object being blurred had a brightness of one everywhere and that the background had a brightness of zero (i.e., the edge of the circle was a step function). Taking the 50% threshold of this linear gradient would place the edge of the motion-blurred particle at the same location as the edge would be had the particle been stationary at its position at the midpoint of the camera exposure period.

After some consideration, we decided it best to simply remove the discussion of particle size measurements as we feel a proper test of the effects of motion blurring on particle size measurements will a) likely require considerable additional work beyond what has already been done prior to and during this review process and b) probably strongly benefit from empirical testing using the actual instruments involved. As the bulk of this paper was focused on the performance of the particle shape measurements, we feel that there is still considerable merit to completing this review process after removing the discussion of particle size measurements. Since the portions of the manuscript to which comments 1 and 2 refer have been removed from the manuscript, we will forgo responding to them. Our response to comment 3 is included below, however.

In addition to removing text relating to the size measurements and changes made in response to comment 3, the other main change to the text is in the methods section, where we have updated the text to reflect that PIP image processing is now applied to the raw PIP images prior to applying the shape-fitting algorithms. This change had been made to the code during a previous round of revisions, but, for whatever reason, the text was not updated to reflect the change.

Lines 184 – 202:
"For the emulation of both MASC and 2DVD as well as the implementation of the tensor method, the PIP images taken from the AVI files undergo image processing matching that of the PIP (Newman et al, 2009; L. Bliven, 2022, personal communication). First, the 8-bit image is passed through a Prewitt edge detection filter and the results are threshold such that all pixels with a value of at least 26 are assigned a value of one and all other pixels are assigned a value of zero. This 2-bit image is then dilated twice using a three-pixel by three-pixel kernel where all elements are set to one. At this point all interior holes are filled with values of one and the resulting 2-bit image is eroded three times using the same kernel as was used in the dilation step. The particles in the resulting image are composed of pixels that have a value of one.

After the image processing, particles are located by matching the particle pixels to the particle positions, as identified in the PIP data files, by using a region grow approach, whereby a coherent region is grown from a single point by checking its eight neighboring pixels for inclusion in the region and then repeating the process with each newly included pixel. The region grow approach is initially attempted at the PIP-

determined particle centroid.  For some particles with large concave regions, the PIP-measured particle center can correspond to a non-particle pixel.  In these instances, an attempt is made to locate a new starting pixel for the region grow technique by searching the following (x,y) coordinate offsets in the given order: (-1,0), (1,0), (0,1), (-1,1), (1,1), (0,-1), (-1,-1), (1,-1).  If a particle pixel is not located at any of these center offset positions, the process is repeated after multiplying the offsets by a factor of two; in cases where a particle pixel remains elusive, the process is repeated with multiples of three and then four.  If no particle pixels are located after checking these 32 offset positions, the particle is considered invalid and is ignored.  Additionally, only particles that appear both in the particle tables (i.e., the PIP files ending in `_a_p.dat') and in one of the two velocity tables (i.e., the PIP files ending in either `_a_v_1.dat' or `_a_v_2.dat') are considered here as the variables contained in these files are necessary when performing the emulations."

Additionally, I've changed the color table for the 2D histogram figures to make them more colorblind friendly (Figs 1, 3, 4, and 8).

**3. Artificial cap and discussions around Fig 7 and Fig 8**

*It is suggested that we perform an additional statistical analysis of the effects of pixilation on the perimeter stretching factor.*

As with many aspects of replicating algorithm functionality in this study, this has become far more complicated and uncertain than it seemed at the outset.  The crux of the issue is in understanding how the IMAQ software package actually computes perimeter.  Again, the frustrations of poor documentation continue.  What documentation does exist for the IMAQ perimeter measurement states that IMAQ subsamples the pixels before measuring the perimeter in order to produce a smoother outline.  Our interpretation of this was that the pixels were subdivided into subpixels and the perimeter was then drawn using the corners of these subpixels, this is apparently wrong though (in hindsight, our interpretation would be better described as oversampling).  I managed to dig up a couple forum posts where users were asking the developer how the algorithm works and the answers, despite still being annoyingly vague, are somewhat illuminating.  Our new understanding is that "subsample" means that IMAQ ignores some pixels along the edge of the particle when drawing the perimeter.  In the example given in one of the forum posts (screenshot and link below as Figure 1), IMAQ skips every other pixel when computing the perimeter and assigns the location of the edge based on one corner of each pixel (how that corner is selected is anyone's guess; my guess is that it is the pixel corner that is farthest from the particle center).  To make things even more troublesome, the number of pixels skipped appears to be a function of the size of the particle; what metric is used for the particle size and how that size relates to the number of pixels skipped is left to our imaginations it would seem.

Regardless, I attempted to test the sensitivity of the perimeter stretching factor to the pixilation of a perfect circle using the path mapped out by IDL's contour procedure as the perimeter.  The circles are generated based on the distance of each pixel center from a randomly perturbed center location and include spatial anti-aliasing to more realistically reflect how a circular particle would appear.  As I was unsure of whether it would be better to have IDL draw the contour at a value of 1.0 or 0.5, I tried both and was troubled to find that there was a fairly large sensitivity to this selection (see Figure 2).  With a value of 1.0, the pixelation effects could be making the perimeter stretching factor smaller or larger, depending on the circle radius.  A value of 0.5, however, suggests the pixelation always makes the perimeter stretching factor larger.  If the results of both were similar to the results using a contour value

of 0.5 (dashed line), I would be more open to agreeing with you that the pixilation was the dominant factor, but given that the 1.0 contour value results in a strong dependence on radius and the vagueness around the subsampling that IMAQ does when computing perimeter, I'm not comfortable with making that statement.  That said, there is a lot more to this than first appeared and I've added some brief text to the manuscript as an explanation.  As for the pixilation vs underlying shape, I've left it as "Small increases in perimeter, such as this, can be introduced by a few very small deviations of the particle edge from a perfect circle as well as by the inability to perfectly represent a circle using square pixels (i.e., pixelation effects)."

[Figure]

*Figure 1.* Screenshot of forum reply explaining IMAQ perimeter calculation.  https://forums.ni.com/t5/Machine-Vision/Vision-perimeter/m-p/1787588#M33709

[Figure]

*Figure 2.* Results of testing pixilation effects on perimeter stretching factor when taking the perimeter to be the IDL contour along a pixel value of 1.0 (solid) and along a pixel value of 0.5 (dashed).

Here are the relevant changes:

Lines 106 – 110:
"Based on online forum discussions[1], it appears that IMAQ computes the particle perimeter from a subset of the edge pixels with the number of edge pixels skipped between each perimeter calculation pixel being a function of particle size. Furthermore, IMAQ uses one corner of each perimeter calculation pixel rather than the pixel center. Unfortunately, we were unable to determine either the function relating particle size to number of skipped pixels or the corner selection method."

The footnote reads: "See the response by user EspenR, an NI application engineer, to the thread "Vision perimeter" on the NI forums: https://forums.ni.com/t5/Machine-Vision/Vision-perimeter/m-p/1787588\#M33709, accessed 29 August 2022."

Lines 350 – 355:
"In theory, it should be possible to determine the contribution of pixelation effects on the PIP shape fitting algorithm; however, as already mentioned, the details of the IMAQ perimeter calculation are not well documented. Additionally, attempting to quantify the contribution from pixelation by measuring the length of a contour around the particle rather than attempting to emulate the IMAQ perimeter calculation resulted in a high sensitivity to the pixel value at which the contour was placed. As such, we have decided to leave the pixelation effects and shape deviations as an unknown in terms of their contributions relative to one another in the PIP shape fitting algorithm."